# Annealing Flow Generative Models Towards Sampling High-Dimensional and Multi-Modal Distributions

## Abstract

Sampling from high-dimensional, multi-modal distributions remains a fundamental challenge across domains such as statistical Bayesian inference and physics-based machine learning. In this paper, we propose *Annealing Flow* (AF), a continuous normalizing flow-based approach designed to sample from high-dimensional and multi-modal distributions. The key idea is to learn a continuous normalizing flow-based transport map, guided by annealing, to transition samples from an easy-to-sample distribution to the target distribution, facilitating effective exploration of modes in high-dimensional spaces. Unlike many existing methods, AF training does not rely on samples from the target distribution. AF ensures effective and balanced mode exploration, achieves linear complexity in sample size and dimensions, and circumvents inefficient mixing times. We demonstrate the superior performance of AF compared to state-of-the-art methods through extensive experiments on various challenging distributions and real-world datasets, particularly in high-dimensional and multi-modal settings. We also highlight AF's potential for sampling the least favorable distributions.

## 1 Introduction

Sampling from high-dimensional and multi-modal distributions is crucial for various fields, including physics-based machine learning like molecular dynamics (Miao et al., 2015; Salo-Ahen et al., 2020), quantum physics (Carlson et al., 2015; Lynn et al., 2019), and lattice field theory (Jay & Neil, 2021; Lozanovski et al., 2020). With modern datasets, it also plays a key role in Bayesian areas, including Bayesian modeling (Balandat et al., 2020; Kandasamy et al., 2018; Stephan et al., 2017) with applications in areas like computational biology (Overstall et al., 2020; Stanton et al., 2022), and Bayesian Neural Network sampling (Cobb & Jalaian, 2021; Izmailov et al., 2021).

*MCMC and Neural Network Variants*: Numerous MCMC methods have been developed over the past 50 years, including Metropolis-Hastings (MH) and its variants (Choi, 2020; Cornish et al., 2019; Griffin & Walker, 2013; Haario et al., 2001), Hamiltonian Monte Carlo (HMC) schemes (Bou-Rabee & Sanz-Serna, 2017; Girolami & Calderhead, 2011; Hoffman et al., 2021; Li et al., 2015; Shahbaba et al., 2014). HMC variants are still considered state-of-the-art methods. However, they require exponentially many steps in the dimension for mixing, even with just two modes (Hackett et al., 2021). More recently, Neural network (NN)-based sampling algorithms (Bonati et al., 2019; Egorov et al., 2024; Gu & Sun, 2020; Hackett et al., 2021; Li et al., 2021; Wolniewicz et al., 2024) have been developed to leverage NN expressiveness for improving MCMC, but they still inherit some limitations like slow mixing and imbalanced mode exploration, particularly in high-dimensional spaces.

*Annealing Variants*: Annealing methods (Gelfand et al., 1990; Neal, 2001; Sorkin, 1991; Van Groenigen & Stein, 1998) are widely used to develop MCMC techniques like Parallel Tempering (PT) and its variants (Chandra et al., 2019; Earl & Deem, 2005; Syed et al., 2022). In annealing, sampling gradually shifts from an easy distribution to the target by lowering temperature. Annealed Importance Sampling (Neal, 2001) and its variants(Chehab et al., 2024; Karagiannis & Andrieu, 2013; Zhang et al., 2021) are developed for estimating normalizing constants with low variance using MCMC samples from intermediate distributions. Recent Normalizing Flow and

score-based annealing methods (Arbel et al., 2021; Doucet et al., 2022) optimize intermediate densities for lower-variance estimates, but still rely on MCMC for sampling. However, MCMC struggles with slow mixing, local mode trapping, mode imbalance, and correlated samples issues. These limitations are particularly pronounced in high-dimensional, multi-modal settings (Hackett et al., 2021; Van Ravenzwaaij et al., 2018).

*Particle Optimization Methods*: Recently, particle-based optimization methods have emerged for sampling, including Stein Variational Gradient Descent (SVGD) (Liu & Wang, 2016), and stochastic approaches such as (Dai et al., 2016; Detommaso et al., 2018; Li et al., 2023; Liu, 2017; Maddison et al., 2018; Nitanda & Suzuki, 2017; Pulido & van Leeuwen, 2019). However, many of these methods rely on kernel computations, which scale polynomially with sample size, and are sensitive to hyperparameters.

*Normalizing Flows*: Recently, Normalizing Flows (NFs) (Rezende & Mohamed, 2015) and Stochastic NFs (Hagemann et al., 2022; Wu et al., 2020) have been explored for sampling. However, discrete NFs often suffer from mode collapse, prompting works (Albergo & Vanden-Eijnden, 2023; Arbel et al., 2021; Brofos et al., 2022; Cabezas et al., 2024; Gabrié et al., 2021; 2022; Matthews et al., 2022) to address this with MCMC corrections, which depend on the quality of MCMC samples and thus may struggle in high-dimensional settings. Several Continuous Normalizing Flows (CNFs) algorithms (Hertrich & Gruhlke, 2024; Tian et al., 2024) are developed to address mode collapse, but still rely on Monte Carlo procedures to correct bias, which are often sensitive to high-dimensional densities. Besides, these methods may often fail with widely-separated modes, leaving some unexplored even after extensive training.

Challenges persist with multi-modal distributions in high-dimensional spaces. This paper introduces *Annealing Flow* (AF), a novel sampling scheme that learns a continuous normalizing flow map from an easy-to-sample distribution $\pi_0(x)$ to the target $q(x)$, guided by annealing principles. Unlike diffusion sampling (Bruna & Han, 2024; Chung et al., 2022; Shih et al., 2024; Zhou et al., 2023) which requires pre-learning from a dataset of unknown distribution, AF training does not require preliminary samples from the target $q(x)$. AF is not based on MCMC, thus avoiding issues like slow mixing, sample correlation, and mode imbalance. And unlike particle-based optimization methods, AF scales linearly with sample size and dimensions. Once trained, one simply samples from $\pi_0(x)$, and the learned transport map directly pushes these samples towards the target distribution.

## 2 PRELIMINARIES

*Neural ODE and Continuous Normalizing Flow:* A Neural ODE is a continuous model where the trajectory of data is modeled as the solution of an ordinary differential equation (ODE). Formally, in $\mathbb{R}^d$, given an input $x(t_0) = x_0$ at time $t_0$, the transformation to the output $x(T)$ is governed by:

$$\frac{dx(t)}{dt} = \mathbf{v}(x(t), t), \tag{1}$$

where $\mathbf{v}(x(t), t)$ represents the velocity field, which is of the same dimension as $x(t)$ and is parameterized by a neural network with input $x(t)$ and $t$.

A Continuous Normalizing Flow (CNF) is a class of normalizing flows where the transformation of a probability density from a base distribution $p(x)$ (at $t = 0$) to a target distribution $q(x)$ (at $t = T$) is governed by a Neural ODE. The marginal density of $x(t)$, denoted as $\rho(x, t)$, evolves according to the continuity equation derived from the ODE in Eq. (1). This continuity equation is written as:

$$\partial_t \rho(x, t) + \nabla \cdot (\rho(x, t)\mathbf{v}(x, t)) = 0, \quad \rho(x, 0) = p(x), \tag{2}$$

where the divergence $\nabla \cdot (\rho v)$ accounts for the change in density as the flow evolves over time.

*Dynamic Optimal Transport (OT):* The Benamou-Brenier equation (Benamou & Brenier, 2000) below provides the dynamic formulation of Optimal Transport $\mathcal{T}$.

$$\inf_{\rho, v} \int_0^1 \mathbb{E}_{x(t) \sim \rho(\cdot, t)} \|\mathbf{v}(x(t), t)\|^2 dt \tag{3}$$
$$\text{s.t.} \quad \partial_t \rho + \nabla \cdot (\rho v) = 0, \quad \rho(\cdot, 0) = p, \quad \rho(\cdot, 1) = q,$$

The optimization problem seeks to find the optimal transport map that moves mass from the base density $p$ to the target density $q$, subject to the continuity equation (2) to ensure that $\rho(\cdot, t)$ evolves as a valid probability density over time. Additionally, the constraint $\rho(\cdot, 1) = q$ ensures that the target density is reached by the end of the time horizon. The time horizon is scaled to $[0, 1]$.

# 3 ANNEALING FLOW MODEL

The annealing philosophy (Gelfand et al., 1990; Neal, 2001; Sorkin, 1991; Van Groenigen & Stein, 1998) refers to gradually transitioning an initial flattened distribution to the target distribution as the temperature decreases. Building on this idea, we introduce Annealing Flow (AF), a sampling algorithm that learns a continuous normalizing flow to gradually map an initial easy-to-sample density $\pi_0(x)$ to the target density $q(x)$ through a set of intermediate distributions.

We define $q(x) = Z\tilde{q}(x)$ where $\tilde{q}(x)$ represents the unnormalized target distribution given in explicit form. Next, we define a sequence of intermediate distributions $f_k(x)$ that interpolate between an easy-to-sample initial distribution $\pi_0(x)$ (e.g., a Gaussian) and the target $q(x)$. These intermediate distributions are formulated as:

$$f_k(x) = \pi_0(x)^{1-\beta_k} q(x)^{\beta_k} = Z_k \tilde{f}_k(x), \tag{4}$$

Here $\tilde{f}_k(x) = \pi_0(x)^{1-\beta_k} \tilde{q}(x)^{\beta_k}$, and $\beta_k$ is an increasing sequence with $\beta_0 = 0$ and $\beta_K = 1$. This formulation ensures that $\tilde{f}_0(x) = \pi_0(x)$ and $\tilde{f}_K(x) = \tilde{q}(x)$. The sequence $0 = \beta_0 < \beta_1 < \cdots < \beta_K = 1$ controls the gradual transition between the two distributions.

The above construction aligns with the annealing philosophy. As $\beta_k$ increases, $\tilde{f}_k(x)$ gradually sharpens toward the target $\tilde{q}(x)$, starting from the initially flattened distribution around $\pi_0(x)$. These annealed densities serve as a bridge, providing a gradual flow path from the easy-to-sample distribution $\pi_0(x)$ to the target density $q(x)$. Figure 1 provides an intuitive illustration of this process, where $\pi_0(x)$ is a standard Gaussian, and $q(x)$ is a Gaussian mixture model with six modes.

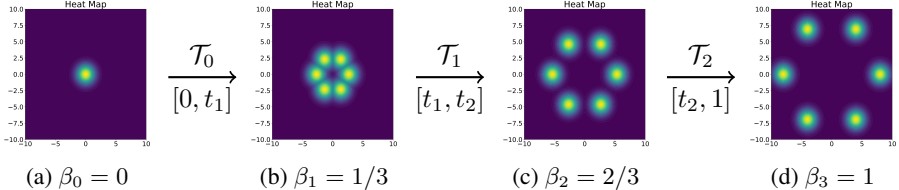

| (a) $\beta_0 = 0$ | (b) $\beta_1 = 1/3$ | (c) $\beta_2 = 2/3$ | (d) $\beta_3 = 1$ |

Figure 1: Illustration of the Annealing Flow Map, with a set of intermediate distributions from $\pi_0(x) = N(0, I_2)$ to $q(x)$, a GMM with 6 modes.

## 3.1 OPTIMAL TRANSPORT MAP

We aim to learn a continuous optimal transport map between an easy-to-sample distribution $\pi_0(x)$ and the target distribution $q(x)$. Once trained, users simply sample $\{x^{(i)}(0)\}_{i=1}^n \sim \pi_0(x)$, and the transport map pushes them to $\{x^{(i)}(1)\}_{i=1}^n \sim q(x)$. The transport map $\mathcal{T}$ evolves the density according to (2), which in turn drives the evolution of the sample $x(t)$ following the ODE in (1):

$$\mathcal{T}(x(t)) = x(0) + \int_0^t \mathbf{v}(x(s), s) ds, \quad t \in [0, 1]. \tag{5}$$

We divide the time horizon $[0, 1]$ of $\mathcal{T}$ into $K$ intervals $[t_{k-1}, t_k]$ for $k = 1, 2, \ldots, K$, where $t_0 = 0$ and $t_K = 1$. Guided by the annealing flow path defined in (4), the continuous flow map $\mathcal{T}$ gradually transforms the density from $f_0(x)$ to $f_1(x)$ over $[0, t_1]$, and continues this process until $f_{K-1}(x)$ is transformed into $f_K(x) = q(x)$ over $[t_{K-1}, t_K]$. Figure 1 shows this progression with two intermediate distributions. For clarity, we denote $\mathcal{T}_k(x)$ as the segment of the continuous normalizing flow during $[t_{k-1}, t_k]$, which pushes the density from $f_{k-1}(x)$ to $f_k(x)$.

## 3.2 Objective of Annealing Flow Net

Annealing Flow aims to learn each transport map $\mathcal{T}_k$ based on dynamic OT objective (3) over the time horizon $[t_{k-1}, t_k]$, where the velocity field $\mathbf{v_k}(x(t), t)$ is learned using a neural network. The terminal condition $\rho(\cdot, 1) = q$ in (3) can be relaxed by introducing a Kullback–Leibler (KL) divergence term (see, for instance, Ruthotto et al. (2020)). Consequently, minimizing the objective (3) for dynamic optimal transport $\mathcal{T}_k : f_{k-1}(x) \to f_k(x)$ can be reduced to solving the following problem:

$$\mathcal{T}_k = \arg\min_{\mathcal{T}} \left\{ \mathrm{KL}(\mathcal{T}_\# f_{k-1} \| f_k) + \gamma \int_{t_{k-1}}^{t_k} \mathbb{E}_{x(t) \sim \rho_k(\cdot, t)} \| \mathbf{v_k}(x(t), t) \|^2 dt \right\}, \tag{6}$$

subject to $\rho_k(x(t), t)$ and $\mathbf{v_k}(x(t), t)$ evolving according to (2). Here, $\gamma > 0$ is a regularization parameter, $\mathbf{v_k}(x(t), t)$ denotes the velocity field during the $k$-th time interval $[t_{k-1}, t_k]$, and $\mathrm{KL}(\mathcal{T}_\# f_{k-1} \| f_k)$ represents the KL divergence between the push-forward density $\mathcal{T}_\# f_{k-1}$ and the target density $f_k$. Additionally, the constraint (2) ensures that $x(t)$ follows the ODE trajectory defined by (1) during $t \in [t_{k-1}, t_k]$, which is given by:

$$x(t) = x(t_{k-1}) + \int_{t_{k-1}}^t \mathbf{v_k}(x(s), s) ds, \quad t \in [t_{k-1}, t_k]. \tag{7}$$

We can rewrite $\tilde{f}_k(x) = Z e^{E_k(x)}$, where $E_k(x)$ is the energy function, with the associated unnormalized energy given by $\tilde{E}_k(x) = -\log \tilde{f}_k$. The following proposition shows that once we have obtained samples from $f_{k-1}(x)$, the KL divergence in (6) can be computed exactly based on $\mathbf{v_k}(x(t), t)$ and $\tilde{E}_k(x)$. Therefore, learning an optimal transport map $\mathcal{T}_k$ reduces to learning the optimal $\mathbf{v_k}(x(t), t)$. The proof is provided in Appendix A.1.

**Proposition 1** (KL-Divergence Decomposition) *Given the unnormalized density $f_{k-1}$, the KL-Divergence between $\mathcal{T}_\# f_{k-1}$ and $f_k$ is equivalent to:*

$$KL(\mathcal{T}_\# f_{k-1} \| f_k) = c + \mathbb{E}_{x(t_{k-1}) \sim f_{k-1}} \left[ \tilde{E}_k(x(t_k)) - \int_{t_{k-1}}^{t_k} \nabla \cdot \mathbf{v_k}(x(s), s) \, ds \right], \tag{8}$$

*up to a constant $c$ that is independent of $\mathbf{v_k}(x(s), s)$.*

Given $x(t_{k-1})$ from $f_{k-1}(x)$, the value of $x(t_k)$ inside the energy function $\tilde{E}_k$ can be calculated as shown in equation (7). Additionally, according to the proposition below, the second term in the objective (6) can be relaxed as a discretized sum. The proof is provided in Appendix A.1.

**Proposition 2** (Wasserstein Distance Discretization) *Let $x(t)$ be particle trajectories driven by a smooth velocity field $\mathbf{v_k}(x(t), t)$ over the time interval $[t_{k-1}, t_k]$, where $h_k = t_k - t_{k-1}$. Assume that $\mathbf{v_k}(x, t)$ is Lipschitz continuous in both $x$ and $t$. By dividing $[t_{k-1}, t_k]$ into $S$ equal mini-intervals with grid points $t_{k-1,s}$ (where $s = 0, 1, \ldots, S$ and $t_{k-1,0} = t_{k-1}$, $t_{k-1,S} = t_k$), we have:*

$$\int_{t_{k-1}}^{t_k} \mathbb{E}_{x(t)} \left[ \| \mathbf{v_k}(x(t), t) \|^2 \right] dt = \frac{S}{h_k} \sum_{s=0}^{S-1} \mathbb{E} \left[ \| x(t_{k-1,s+1}) - x(t_{k-1,s}) \|^2 \right] + O(h_k^2 / S). \tag{9}$$

*As $h_k \to 0$ or $S \to \infty$, the error term $O(h_k^2 / S)$ becomes negligible.*

One can observe that the RHS of (9) can be interpreted as the discretized sum of the squared Wasserstein-2 distance. The dynamic $W_2$ regularization encourages smooth transitions from $f_{k-1}$ to $f_k$ with minimal transport cost, promoting efficient mode exploration.

Next, by incorporating Propositions 1 and 2 into objective (6), the *final objective* becomes:

$$\min_{\mathbf{v_k}(\cdot, t)} \mathbb{E}_{x(t_{k-1}) \sim f_{k-1}} \left[ \tilde{E}_k(x(t_k)) - \int_{t_{k-1}}^{t_k} \nabla \cdot \mathbf{v_k}(x(s), s) ds + \alpha \sum_{s=0}^{S-1} \| x(t_{k-1,s+1}) - x(t_{k-1,s}) \|^2 \right]. \tag{10}$$

Here, $\alpha = \gamma S / h_k$ and $\mathbf{v_k}(x(s), s)$ is learned by a neural network. We break the time interval $[t_{k-1}, t_k]$ into $S$ mini-intervals, and $x(t_{k-1,s+1})$ is computed as in equation (7).

After learning, connecting the Annealing Flow nets together yields a smooth flow map $\mathcal{T} : \mathcal{T}_1 \to \mathcal{T}_2 \to \cdots \to \mathcal{T}_K$, which transforms samples from $\pi_0(x)$ to the target $q(x)$. Please see Section 4.2 for efficient sampling of Annealing Flow and its comparisons with other sampling methods.

### 3.3 PROPERTIES OF LEARNED VELOCITY FIELD

The objective in (10) can be reformulated as shown below when $h_k = t_k - t_{k-1} \to 0$. The proof is provided in Appendix A.2.

**Proposition 3** (Objective Reformulation) *Denote $h_k = t_k - t_{k-1}$, and let $\mathbf{s_k} = \nabla \log f_k(x)$ denote the score function of $f_k$. As $h_k \to 0$ and with $\gamma = \frac{1}{2}$ (so that $\alpha = \frac{S}{2h_k}$), the objective in (10) becomes equivalent to the following:*

$$\min_{\mathbf{v_k} = \mathbf{v_k}(\cdot, 0)} \mathbb{E}_{x \sim f_{k-1}} \left[ -T_{f_k} \mathbf{v_k} + \frac{1}{2} \|\mathbf{v_k}\|^2 \right], \quad T_{f_k} \mathbf{v_k} := \mathbf{s_k} \cdot \mathbf{v_k} + \nabla \cdot \mathbf{v_k}. \tag{11}$$

Define $L^2(f_{k-1}) = \left\{ v : \mathbb{R}^d \to \mathbb{R}^d \mid \int_{\mathbb{R}^d} \|\mathbf{v}(x)\|^2 f_{k-1}(x)\, dx < \infty \right\}$ as the $L^2$ space over $(\mathbb{R}^d, f_{k-1}(x)dx)$. We can then establish the following property, with proofs provided in Appendix A.2:

**Proposition 4** (Optimal Velocity Field as Score Difference) *Suppose $h_k \to 0$ and $\gamma = \frac{1}{2}$. Let $f_{k-1}$ and $f_k$ be continuously differentiable on $\mathbb{R}^d$. Assume that $\nabla \cdot \mathbf{v_k}(x)$ exists for all $x \in \mathbb{R}^d$, and $\nabla \cdot \mathbf{v_k}(x)$, $\mathbf{s_{k-1}}$ and $\mathbf{s_k}$ belong to $L^2(f_{k-1})$. Assume that the components of $\mathbf{v_k}$ are independent and $\lim_{\|x\| \to \infty} f_{k-1}(x) \|\mathbf{v_k}(x)\|_2 = 0$. Under these conditions, the minimizer of (10) is:*

$$\mathbf{v_k}^* = \mathbf{s_k} - \mathbf{s_{k-1}}. \tag{12}$$

Therefore, the infinitesimal optimal $\mathbf{v_k}^*$ is equal to the difference between score function of the next density, $f_k$, and the current density, $f_{k-1}$. This suggests that when the two intermediate densities are sufficiently close, i.e., when the number of $\beta_k$ is large enough, the optimal velocity field equals the difference between the score functions. By adding more intermediate densities, one can construct a sufficiently smooth transport map $\mathcal{T}$ that exactly learns the mapping between each pair of densities.

Additionally, one can observe that when each $\tilde{f}_k(x)$ is set to the target $q(x)$, i.e., when all $\beta_k$ are set to 1, and the second term in the objective (6) is relaxed to static $W_2$ regularization, the objective of Annealing Flow becomes equivalent to Wasserstein gradient flow. This is detailed in Appendix B.

## 4 TRAINING AND SAMPLING OF ANNEALING FLOW NET

### 4.1 BLOCK-WISE TRAINING

Training of the $k$-th flow map in Annealing Flow begins once the $(k-1)$-th block has completed training. Given the samples $\{x^{(i)}(t_{k-1})\}_{i=1}^n \sim f_{k-1}(x)$ produced after the $(k-1)$-th block, we can replace $\mathbb{E}_{x \sim f_{k-1}}$ with the empirical average. The divergence of the velocity field can be computed either by brute force or via the Hutchinson trace estimator (Hutchinson, 1989; Xu et al., 2024a):

$$\nabla \cdot \mathbf{v_k}(x, t) \approx \mathbb{E}_{\epsilon \sim N(0, I_d)} \left[ \epsilon^T \frac{\mathbf{v_k}(x + \sigma \epsilon, t) - \mathbf{v_k}(x, t)}{\sigma} \right]. \tag{13}$$

This approximation becomes exact as $\sigma \to 0$. Further details are provided in C.2. Additionally, we apply the Runge-Kutta method for numerical integration, with details provided in C.3.

Our algorithm uses a block-wise training of the continuous normalizing flow map. Specifically, the training of Annealing Flow is summarized in Algorithm 1. The block-wise training approach of Annealing Flow significantly reduces memory and computational requirements, as only one neural network is trained at a time, independent of the other flow networks.

### 4.2 EFFICIENT SAMPLING AND COMPARISONS WITH OTHER METHODS

Once the continuous normalizing flow map $\mathcal{T}$ is learned, the sampling process of the target $q(x)$ can be very efficient. Users can simply sample $\{x^{(i)}(t_0 = 0)\}_{i=1}^n$ from $\pi_0(x)$, and then directly calculate $\{x^{(i)}(t_K = 1)\}_{i=1}^n \sim q(x)$ through Annealing Flow nets:

$$x^{(i)}(t_k) = \mathcal{T}_k(x^{(i)}(t_{k-1})) = x^{(i)}(t_{k-1}) + \int_{t_{k-1}}^{t_k} \mathbf{v_k}(x^{(i)}(s), s)ds, \quad k = 1, 2, \cdots, K. \tag{14}$$

---

**Algorithm 1** Block-wise Training of Annealing Flow Net

---

**Require:** Unnormalized target density $\tilde{q}(x)$; an easy-to-sample $\pi_0(x)$; $\{\beta_1, \beta_2, \cdots, \beta_{K-1}\}$; Total number of blocks $K$.

1: Set $\beta_0 = 0$ and $\beta_K = 1$

2: For $k = 1, 2, \cdots, K$:

3:     Set $\tilde{f}_k(x) = \pi_0(x)^{1-\beta_k}\tilde{q}(x)^{\beta_k}$;

4:     Sample $\{x^{(i)}(t_0)\}_{i=1}^n$ from $\pi_0(x)$;

5:     Compute the pushed samples $x^{(i)}(t_{k-1})$ from the trained $(k-1)$ blocks via (14);

6:     Optimize $\mathbf{v_k}(\cdot, t)$ upon minimizing the objective function.

   (Optional Refinement Blocks)

7: For $k = K+1, K+2, \cdots, L$:

8:     Set $\beta_k = 1$ and optimize $\mathbf{v_k}(\cdot, t)$ following the procedures outlined above.

---

MCMC methods require long mixing times when sampling from complex distributions. In contrast, Annealing Flow (AF) pushes samples directly from $\pi_0(x)$ through the learned transport map, enabling faster sampling, especially for large sample sizes. MCMC also generates correlated samples, as each new sample depends on the previous one, reducing the effective sample size (ESS) and efficiency. AF avoids this by producing independent samples, improving overall sample quality.

Additionally, MCMC struggles with multimodal distributions, as chains get trapped in local modes. While methods like Parallel Tempering may attempt to explore all modes in low-dimensions, they do not ensure proportional time across them, causing imbalanced sampling. In contrast, AF generates balanced samples across modes in line with the target distribution, as illustrated in the below figure.

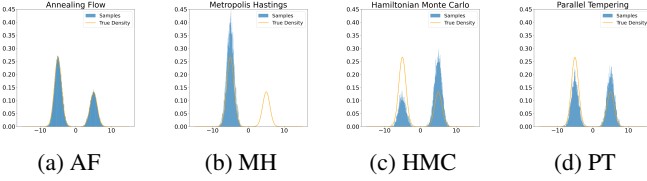

(a) AF      (b) MH      (c) HMC      (d) PT

Figure 2: Comparison of different sampling methods for the density $p(x) = \frac{2}{3}N(-5, 1) + \frac{1}{3}N(5, 1)$

NN-based MCMC algorithms still struggle with issues like slow mixing and correlated samples. Particle-based methods like SVGD and MIED avoid burn-in period and produce less correlated samples, but their reliance on kernel computations leads to polynomial scaling with sample size, and they are sensitive to kernel hyperparameters. In contrast, AF computes samples independently through (14), allowing the sampling process to scale linearly with both sample size and dimensions.

We comment that Annealing Flow indeed needs more expensive pre-training than MCMC, which, however, can be done offline and only needs to be done once and then deployed for sampling. Once trained, AF samplers are highly efficient, generating 10,000 samples in an average of 1.5 seconds in our experiments. In contrast, MCMC takes around 1 minute to sample 10,000, while particle-based methods take significantly longer—over 20 minutes. AF also performs well on multimodal and high-dimensional densities, where other methods often struggle. Detailed comparisons of algorithms, including the training and sampling times, are provided in D.1.

## 5   IMPORTANCE FLOW

Sampling from complex distributions is fundamental, which can benefit tasks like normalizing constant estimation, Bayesian analysis, and various machine learning problems. Here, we briefly discuss another aspect: using Annealing Flow to sample from the Least-Favorable-Distribution (LFD) and obtain a low-variance Importance Sampling (IS) estimator, referred to as Importance Flow.

## 5.1 SETTINGS

Suppose we want to estimate $\mathbb{E}_{X \sim \pi_0(x)}[h(X)]$, which cannot be computed in closed form. A natural approach is to use Monte Carlo estimation by sampling $\{x_i\}_{i=1}^n$ from $\pi_0(x)$. However, if $x_i$ consistently falls in regions where $h(x)$ has extreme values, the estimator may exhibit high variance. For example, with $\pi_0(x) = N(0, I_d)$ and $h(x) = 1_{\|x\| \geq 6}$, almost no samples will satisfy $\|x\| \geq 6$, resulting in a zero estimate.

To address this situation, we can select an appropriate proposal distribution $q(x)$ and rewrite the expectation and MC estimator as:

$$\mathbb{E}_{x \sim \pi_0(x)}[h(x)] = \mathbb{E}_{x \sim q(x)}\left[\frac{\pi_0(x)}{q(x)}h(x)\right] \approx \frac{1}{n}\sum_{i=1}^n \frac{\pi_0(x_i)}{q(x_i)}h(x_i), \quad x_i \sim q(x). \tag{15}$$

It is well-known that the theoretically optimal proposal for the importance sampler is: $q^*(x) \propto \pi_0(x)|h(x)| := \tilde{q}^*(x)$. However, given the definition of $\tilde{q}^*(x)$, it is often difficult to sample from, especially when $\pi_0(x)$ or $h(x)$ is complex. Consequently, people typically choose a distribution that is similar in shape to the theoretically optimal proposal but easier to sample from.

Annealing Flow enables sampling from $q^*(x)$, allowing the construction of an Importance Sampling (IS) estimator. However, $q^*(x)$ is only known up to the normalizing constant $Z$, where $q^*(x) = \frac{1}{Z}\tilde{q}(x)$ and $Z = \mathbb{E}_{x \sim \pi_0(x)}[h(x)]$ is our target. Therefore, assuming no knowledge on $Z$, a common choice can be the Normalized IS Estimator: $\hat{I}_N = \sum_{i=1}^n \frac{\pi_0(x_i)}{\tilde{q}(x_i)}h(x_i) / \sum_{i=1}^n \frac{\pi_0(x_i)}{\tilde{q}(x_i)}$. However, this estimator is often biased, as can be seen from Jensen's Inequality.

## 5.2 DENSITY RATIO ESTIMATION

Using samples from $q^*(x)$ and those along the trajectory obtained via Annealing Flow, we can train a neural network for Density Ratio Estimation (DRE) of $\frac{\pi_0(x)}{q^*(x)}$. Inspired by works Choi et al. (2022); Rhodes et al. (2020); Xu et al. (2023), we can train a continuous neural network $r(x) = r_K(x; \theta_K) \circ r_{K-1}(x; \theta_{K-1}) \circ \cdots \circ r_1(x; \theta_1)$, where samples $x_i \sim f_K = q^*(x)$ are inputs and the output is the density ratio $\frac{\pi_0(x_i)}{q^*(x_i)}$. Each $r_k(x; \theta_k)$ is trained using the following loss:

$$\mathcal{L}_k(\theta_k) = \mathbb{E}_{x(t_{k-1}) \sim f_{k-1}}\left[\log(1 + e^{-r_k(x_i(t_{k-1}))})\right] + \mathbb{E}_{x(t_k) \sim f_k}\left[\log(1 + e^{r_k(x_i(t_k))})\right].$$

After successful training, $r_k^*(x) = \log\frac{f_{k-1}(x)}{f_k(x)}$, and thus $r^*(x) = \sum_{k=1}^K r_k^*(x) = \log\frac{\pi_0(x)}{q^*(x)}$. Please refer to Appendix A.3 and C.5 for the proof and further details. To obtain the optimal importance sampling estimator, we can then directly use samples $\{x_i\}_{i=1}^n \sim q^*(x)$ from Annealing Flow and apply (15) together with the DRE: $\frac{1}{n}\sum_{i=1}^n \exp(r^*(x_i)) \cdot h(x_i)$. The estimator is unbiased and can achieve zero variance theoretically.

## 6 NUMERICAL EXPERIMENTS

In this section, we present numerical experiments comparing Annealing Flow (AF) with widely-used MCMC algorithms, including Hamiltonian Monte Carlo (HMC) and Parallel Tempering (PT), as well as other state-of-the-art techniques, including particle-based methods: Stein Variational Gradient Descent (SVGD) (Liu & Wang, 2016) and Mollified Interaction Energy Descent (MIED) (Li et al., 2023), alongside NN-based MCMC approaches: AI-Sampler (AIS) (Egorov et al., 2024). The experimental details can be found in C.3.

We test these algorithms on challenging distributions, including Exp-Weighted Gaussian, Gaussian Mixture Models (GMM), funnel distributions, and Truncated Normal with extreme radii across varying dimensions. Maximum Mean Discrepancy (MMD) and Wasserstein Distance are used as evaluation metrics, but only reported for the GMM due to the need for true samples. For other experiments, we provide sample and density plots for easier comparison, as shown in Appendix D.

In addition, we compare our algorithm with others on Hierarchical Bayesian Logistic Regression across a range of datasets. We also report the preliminary results of the Importance Flow (discussed in Section 5) for estimating $\mathbb{E}_{x \sim N(0, I)}[1_{\|x\| \geq c}]$ with varying $c$ and dimensions.

*Gaussian Mixture Models (GMM):* Figure 3 presents the sampling results of different methods on a $2D$ GMM, where the modes are distributed across circles with varying radii. We also experimented on a GMM with modes aligned on the vertices of a cube in higher dimensions, with the number of modes ranging from 8 to *64*. Evaluation metrics and additional figures for these experiments are provided in Table 4 in Appendix D.

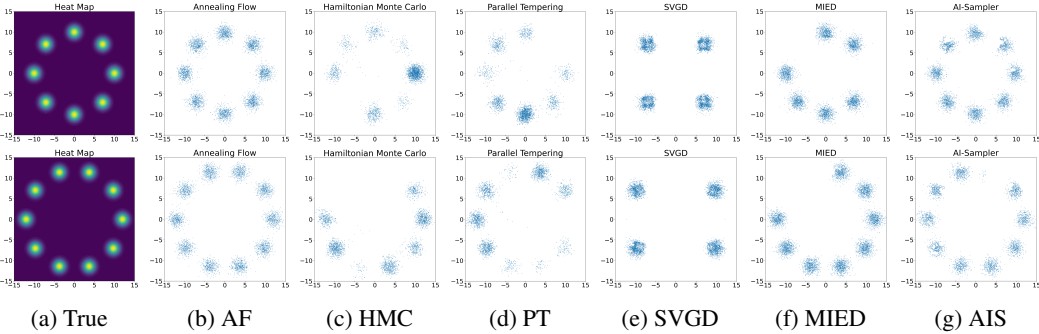

| (a) True | (b) AF | (c) HMC | (d) PT | (e) SVGD | (f) MIED | (g) AIS |

Figure 3: Sampling methods for Gaussian Mixture Models (GMM) with 8 and 10 modes distributed on circles with radii $r = 10, 12$. The acronyms of the methods are listed in the first paragraph of this section.

*Truncated Normal Distribution:* Figure 4 shows the sampling results for $\tilde{q}(x) = 1_{\|x\| \geq c} N(0, I_d)$, based on 5000 samples for each method. SVGD, MIED, and AI-Sampler are designed for continuous densities. SVGD and MIED specifically require the gradient of the log-probability, given by $\nabla \log \left( 1_{\|x\| \geq c} N(0, I_d) \right)$ in this experiment. Despite relaxing the indicator function to $1/(1 + \exp(-k(\|x\| - c)))$ for large $k$, the algorithms failed to yield meaningful results (See Figure 8 in Appendix D for the results of their algorithms). Therefore, we compare AF with MH, HMC, and PT. We also tested our algorithm on *10D* space. Additional figures are given in Appendix D.

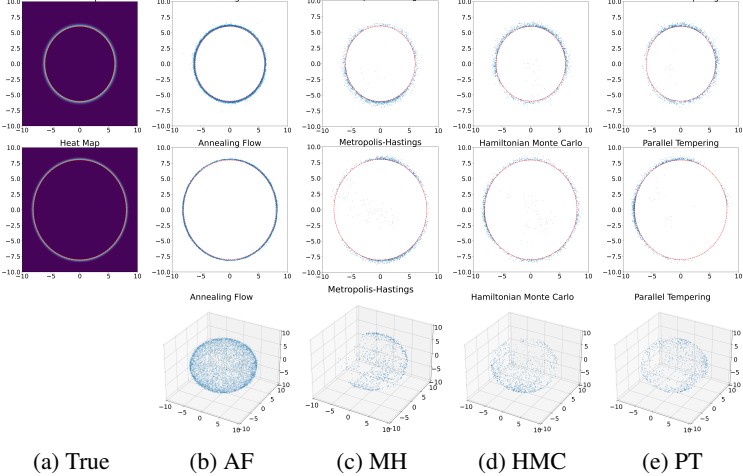

| (a) True | (b) AF | (c) MH | (d) HMC | (e) PT |

Figure 4: Sampling methods for truncated normal distributions with radii $c = 6$ and $c = 8$ in 2D space for the first two rows. The last row presents sampling results in *5D* with a radius of 8, projected onto a $3D$ space.

*Funnel Distribution:* A well-known challenging distribution for sampling is the funnel distribution, defined as:

$$P(x_1, x_2, \ldots, x_d) \propto \mathcal{N}(x_1 \mid 0, \sigma^2) \prod_{i=2}^{d-1} \mathcal{N}(x_i \mid 0, \exp(x_1)),$$

In this setup, $x_i, i = 2, \cdots, d$ has a variance that depends exponentially on $x_1$, forming a funnel-shaped distribution. Sampling is challenging due to this exponential dependence, causing extreme concentration for negative $x_1$ and wide dispersion for positive $x_1$, making exploration difficult, especially in high dimensions.

We tested our Annealing Flow together with other algorithms on $d = 5$ case. Here, we present the sampling result projected onto a $3D$ space for a funnel distribution in a $5D$ space, with $\sigma^2 = 0.81$:

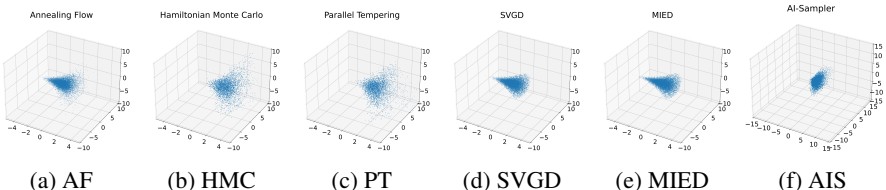

| (a) AF | (b) HMC | (c) PT | (d) SVGD | (e) MIED | (f) AIS |

Figure 5: Sampling Methods for Funnel Distribution in $d = 5$, projected onto $d = 3$.

*Exp-Weighted Gaussian with an Extreme Number of Modes in High-Dimensional Spaces:*

We tested each algorithm on sampling from an extreme distribution:

$$p(x_1, x_2, \cdots, x_{10}) \propto e^{10 \sum_{i=1}^{10} |x_i| - \frac{1}{2} \|x\|^2},$$

which has $2^{10} = 1024$ modes arranged at the vertices of a *10-D cube*. The L2-distance between two horizontally or vertically adjacent modes is 20, while the diagonal modes are separated by up to $\sqrt{10 \cdot 20^2} \approx 63.25$. We also tested on the extreme distribution:

$$p(x_1, x_2, \cdots, x_{50}) \propto e^{10 \sum_{i=1}^{10} |x_i| + 10 \sum_{i=11}^{50} x_i - \frac{1}{2} \|x\|^2},$$

which has $2^{10} = 1024$ modes arranged at the vertices of a *50-D space*.

Given the challenge of visualizing results in high-dimensional space, we present in Figure 6 the projected results of the *50-D* samples onto the first three dimensions. For comparisons in *10-D* space, please refer to Figure 12 in Appendix D. The performance of SVGD, MIED, and AIS is inferior to AF, as compared in Figures 13 and 14 in Appendix D.

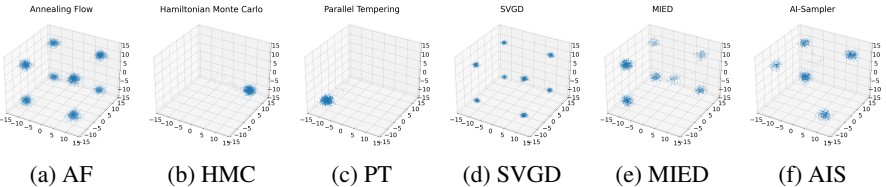

| (a) AF | (b) HMC | (c) PT | (d) SVGD | (e) MIED | (f) AIS |

Figure 6: Sampling Methods for an Exp-Weighted Gaussian Distribution with 1024 modes in Dimension $d = 50$, projected onto a $d = 3$ Space.

Table 1: The number of modes successfully explored by each algorithm across various dimensions.

| | $d = 2$ | $d = 5$ | $d = 10$ | $d = 50$ | | $d = 2$ | $d = 5$ | $d = 10$ | $d = 50$ |
|---|---|---|---|---|---|---|---|---|---|
| True | 4 | 32 | 1024 | 1024 | SVGD | 3.9 | 28.5 | 957.3 | 916.4 |
| **AF** | **4** | **32** | **1024** | **1024** | MIED | 3.8 | 28.0 | 923.4 | 890.6 |
| HMC | 3.1 | 24.3 | 213.5 | $< 10$ | AIS | 3.8 | 28.3 | 707.4 | 125.6 |
| PT | 3.4 | 25.2 | 233.7 | $< 10$ | | | | | |

Table 1 presents the number of modes successfully explored by different algorithms across varying dimensions. Each algorithm was run 10 times, sampling 10,000 points per run, and the average number of modes explored by each algorithm was then calculated.

*Bayesian Logistic Regression:* We use the same Bayesian logistic regression setting as in Liu & Wang (2016), where a hierarchical structure is assigned to the model parameters. The weights $\beta$ follow a Gaussian prior $p_0(\beta|\alpha) = N(\beta; 0, \alpha^{-1})$, and $\alpha$ follows a Gamma prior $p_0(\alpha) =$ Gamma$(\alpha; 1, 0.01)$. Sampling is performed on the posterior $p(\beta, \alpha|D)$, where $D = \{x_i, y_i\}_{i=1}^n$. The performance comparisons are shown in Table 2. Detailed settings are given in C.4.

Table 2: Bayesian Logistic Regression: comparison of different algorithms across datasets. In the table $\cdot \pm \cdot/\cdot$ represents Accuracy(%)±std(%)/log-posterior

| Dataset | AF | SVGD | MIED | AI-Sampler |
|---|---|---|---|---|
| Diabetes ($d = 8$) | **76.30 ± 2.12**/ − 0.496 | 76.10 ± 2.5/ − 0.502 | 75.80 ± 2.32/ − 0.503 | **76.30 ± 2.18**/−0.493 |
| Breast Cancer ($d = 10$) | 97.85 ± 1.12/ − 0.017 | **98.83 ± 3.10**/−0.008 | **98.89 ± 2.12**/−0.008 | 97.83 ± 2.80/ − 0.019 |
| Heart ($d = 13$) | **88.46 ± 2.73**/−0.316 | 79.36 ± 3.78/ − 0.588 | 86.70 ± 2.24/ − 0.321 | 84.23 ± 2.54/ − 0.458 |
| Australian ($d = 14$) | **86.59 ± 1.20**/−0.361 | 84.56 ± 2.87/ − 0.365 | 85.17 ± 1.34/ − 0.369 | 84.62 ± 2.30/ − 0.375 |
| Ijcnn1 ($d = 22$) | **91.96 ± 0.05**/−0.195 | 89.44 ± 0.34/ − 0.209 | 91.84 ± 0.15/−0.198 | 88.32 ± 0.25/ − 0.334 |
| Svmguide3 ($d = 22$) | 80.04 ± 0.70/ − 0.472 | 78.89 ± 1.20/ − 0.479 | 80.12 ± 1.04/−0.472 | 80.12 ± 0.98/−0.468 |
| German ($d = 24$) | **78.04 ± 1.70**/−0.473 | 76.43 ± 1.70/ − 0.483 | 77.21 ± 1.80/ − 0.479 | 76.89 ± 1.84/ − 0.484 |

*Importance Flow:* Table 3 reports the preliminary results of the importance flow (discussed in Section 5) for estimating $\mathbb{E}_{x \sim N(0,I)}\left[1_{\|x\| \geq c}\right]$ with varying radii $c$ and dimensions. This estimation uses samples from the experiment on the Truncated Normal Distribution, and thus the results for SVGD, MIED, and AIS cannot be reported. Please refer to C.5 for detailed experimental settings. Additionally, we discussed a possible extension of the Importance Flow framework in D.2.

Table 3: Comparison of Results for different radii (c) and dimensions (d). The value in parentheses indicates the standard deviation.

| Methods | Radius | $d = 2$ | $d = 3$ | $d = 4$ | $d = 5$ |
|---|---|---|---|---|---|
| True Probability | $c = 4$ | 3.35e-04 | 1.13e-03 | 3.02e-03 | 6.84e-03 |
| | $c = 6$ | 1.52e-08 | 7.49e-08 | 2.89e-07 | 9.50e-07 |
| Importance Flow | $c = 4$ | **4.04e-04(1.0e-04)** | 1.30e-03(**2.3e-04**) | **3.36e-03(4.23e-04)** | **7.86e-03(8.21e-04)** |
| | $c = 6$ | **9.81e-08(4.02e-07)** | **1.51e-07(1.23e-07)** | **2.13e-07(8.71e-08)** | **2.38e-07(3.48e-06)** |
| DRE with HMC Samples | $c = 4$ | 7.56e-04(4.99e-04) | 2.52e-03(6.33e-04) | 8.97e-03(9.05e-04) | 1.12e-02(1.55e-03) |
| | $c = 6$ | 4.35e-07(7.21e-07) | 9.01e-07(2.79e-06) | 1.82e-07(2.89e-06) | 2.31e-06(6.21e-06) |
| DRE with PT Samples | $c = 4$ | 6.79e-04(3.58e-04) | 2.38e-03(5.40e-04) | 5.78e-03(7.98e-03) | 9.94e-03(1.13e-03) |
| | $c = 6$ | 5.37e-07(9.56e-07) | 8.78e-07(2.32e-06) | 9.23e-07(2.51e-06) | 1.98e-06(7.73e-06) |
| Naïve MC | $c = 4$ | 2.75e-04(6.0e-04) | **1.18e-03**(1.1e-03) | **2.71e-03**(1.7e-03) | 7.94e-03(2.6e-03) |
| | $c = 6$ | 0 | 0 | 0 | 0 |

# 7 DISCUSSIONS

In this paper, we have proposed the Annealing Flow (AF) framework, a novel and flexible approach for sampling from high-dimensional and multi-modal distributions. AF offers several advantages over existing methods, as thoroughly discussed in D.1. Additionally, we have also compared the training and sampling times in D.1. Extensive experiments demonstrate that AF performs well across a variety of challenging distributions and real-world datasets.

The Annealing Flow framework presented in this paper is highly flexible and accommodates various challenging distributions. The concept of 'Annealing' in sampling can be interpreted as gradually transitioning from an easy-to-sample distribution to the target distribution. Therefore, each intermediate distribution $f_k$ can be defined flexibly without adhering to (4), as long as the transitions between $f_{k-1}$ and $f_k$ are smooth and the sequence converges to the target $q(x)$. If the density modes are close enough, all $\tilde{f}_k(x)$ can simply be set to the target density $q(x)$, making the Annealing Flow objective equivalent to the Wasserstein gradient flow, as discussed in Appendix B. Additionally, we believe that by adding more intermediate distributions, one can obtain intermediate samples at various time points to construct a low-variance estimator for the normalizing constant. Finally, the importance flow discussed in Section 5 may be extended to a distribution-free model, allowing one to learn an importance flow from a dataset for sampling its Least-Favorable Distribution (LFD) with minimal variance, as further detailed in D.2.

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

# A   PROOFS

## A.1   PROOFS IN SECTION 3.2

**Proposition 1.** (KL-Divergence Decomposition) *Given the unnormalized density $f_{k-1}$, the KL-Divergence between $\mathcal{T}_\# f_{k-1}$ and $f_k$ is equivalent to:*

$$KL(\mathcal{T}_\# f_{k-1} \| f_k) = c + \mathbb{E}_{x \sim f_{k-1}} \left[ \tilde{E}_k(x(t_k)) - \int_{t_{k-1}}^{t_k} \nabla \cdot \mathbf{v_k}(x(s), s) \, ds \right],$$

*up to a constant $c$ that is independent of $\mathbf{v_k}(x(s), s)$.*

*Proof:*

Let $\rho(x, t)$ denote the density evolution under the transport map $\mathcal{T}$, as defined in (2). By the constraint (2) in the transport map objective (3), we have $\mathcal{T}_\# f_{k-1}(x) = \rho(x, t_k)$. The expression for KL-divergence is given by:

$$KL(\mathcal{T}_\# f_{k-1} \| f_k) = \mathbb{E}_{x \sim \rho(x, t_k)} \left[ \log \frac{\mathcal{T}_\# f_{k-1}(x)}{f_k(x)} \right] = \mathbb{E}_{x \sim \rho(x, t_k)} \left[ \log \mathcal{T}_\# f_{k-1}(x) - \log f_k(x) \right].$$

Now, recall that $-\log \tilde{f}_k(x) = \tilde{E}_k(x)$, so we substitute:

$$KL(\mathcal{T}_\# f_{k-1} \| f_k) = \mathbb{E}_{x \sim \rho(x, t_k)} \left[ \log \mathcal{T}_\# f_{k-1}(x) + \tilde{E}_k(x) \right] - \log Z_k$$

$$= \mathbb{E}_{x \sim \rho(x, t_{k-1})} \left[ \log T_\# f_{k-1}(x(t_k)) + \tilde{E}_k(x(t_k)) \right] - \log Z_k,$$

where the second equality holds under the constraints (1) and (2). The density $\rho$ evolves according to (2), and equivalently, the particles $x(t)$ evolve according to (1).

Next, to compute $\log T_\# f_{k-1}(x(t_k))$, we use the fact that the dynamics of the pushforward density $\rho$ are governed by the velocity field $\mathbf{v_k}(x(s), s)$:

$$\frac{d}{ds} \log \rho(x(s), s) = \frac{\nabla \rho(x(s), s) \cdot \partial_s x(s) + \partial_s \rho(x(s), s)}{\rho(x(s), s)}$$

$$= \left. \frac{\nabla \rho \cdot \mathbf{v_k} - \nabla \cdot (\rho \mathbf{v_k})}{\rho} \right|_{(x(s), s)} \quad \text{(by (1) and (2))}$$

$$= \left. \frac{\nabla \rho \cdot \mathbf{v_k} - (\nabla \rho \cdot \mathbf{v_k} + \rho \nabla \cdot \mathbf{v_k})}{\rho} \right|_{(x(s), s)}$$

$$= -\nabla \cdot \mathbf{v_k}(x(s), s).$$

Integrating this equation over the interval $s \in [t_{k-1}, t_k]$, we find:

$$\log \mathcal{T}_\# f_{k-1}(x(t_k)) = \log \rho(x(t_k), t_k) = \log \rho(x(t_{k-1}), t_{k-1}) - \int_{t_{k-1}}^{t_k} \nabla \cdot \mathbf{v_k}(x(s), s) ds.$$

We now substitute this result back into the KL-divergence expression:

$$KL(\mathcal{T}_\# f_{k-1} \| f_k) = \mathbb{E}_{x \sim \rho(x, t_{k-1})} \left[ \log \rho(x(t_{k-1}), t_{k-1}) - \int_{t_{k-1}}^{t_k} \nabla \cdot \mathbf{v_k}(x(s), s) ds + \tilde{E}_k(x(t_k)) \right] - \log Z_k.$$

Note that $\mathbb{E}_{x \sim \rho(x(t_{k-1}), t_{k-1})} \left[ \log \rho(x(t_{k-1}), t_{k-1}) \right]$ is independent of $\mathbf{v_k}(x(s), s)$ and thus acts as a constant term, along with $-\log Z_k$, which we now denote as $c$. After successfully training the previous velocity fields, we have $\rho(x, t_{k-1}) = f_{k-1}(x)$. Therefore, the relevant terms for the KL-divergence are:

$$KL(\mathcal{T}_\# f_{k-1} \| f_k) = c + \mathbb{E}_{x \sim f_{k-1}} \left[ \tilde{E}_k(x(t_k)) - \int_{t_{k-1}}^{t_k} \nabla \cdot \mathbf{v_k}(x(s), s) ds \right].$$

**Proposition 2.** (Wasserstein Distance Discretization) *Let $x(t)$ be particle trajectories driven by a smooth velocity field $\mathbf{v_k}(x(t), t)$ over the time interval $[t_{k-1}, t_k]$, where $h_k = t_k - t_{k-1}$. Assume that $\mathbf{v_k}(x, t)$ is Lipschitz continuous in both $x$ and $t$. By dividing $[t_{k-1}, t_k]$ into $S$ equal mini-intervals with grid points $t_{k-1,s}$ (where $s = 0, 1, \ldots, S$ and $t_{k-1,0} = t_{k-1}$, $t_{k-1,S} = t_k$), the following approximation holds:*

$$\int_{t_{k-1}}^{t_k} \mathbb{E}_{x(t)} \left[ \|\mathbf{v_k}(x(t), t)\|^2 \right] dt = \frac{S}{h_k} \sum_{s=0}^{S-1} \mathbb{E} \left[ \|x(t_{k-1,s+1}) - x(t_{k-1,s})\|^2 \right] + O\left(h_k^2/S\right).$$

*As $h_k \to 0$ or $S \to \infty$, the error term $O\left(h_k^2/S\right)$ becomes negligible.*

*Proof:*

Consider particle trajectories $x(t)$ driven by a sufficiently smooth velocity field $\mathbf{v_k}(x(t), t)$ over the time interval $[t_{k-1}, t_k]$, where $h_k = t_k - t_{k-1}$. We divide this interval into $S$ equal mini-intervals of length $\delta t = \frac{h_k}{S}$, resulting in grid points $t_{k-1,s} = t_{k-1} + s\delta t$ for $s = 0, 1, \ldots, S$, where $\delta t = \frac{t_k - t_{k-1}}{S}$.

Within each mini-interval $[t_{k-1,s}, t_{k-1,s+1}]$, we perform a Taylor expansion of $x(t)$ around $t_{k-1,s}$:

$$x(t_{k-1,s+1}) = x(t_{k-1,s}) + \mathbf{v_k}(x(t_{k-1,s}), t_{k-1,s})\delta t + \frac{1}{2}\frac{d\mathbf{v_k}}{dt}\delta t^2 + O(\delta t^3),$$

where $\frac{d\mathbf{v_k}}{dt}$ denotes the total derivative of $\mathbf{v_k}$ with respect to time.

The squared displacement over the mini-interval $[t_{k-1,s}, t_{k-1,s+1}]$ is given by:

$$\|x(t_{k-1,s+1}) - x(t_{k-1,s})\|^2 = \left\| \mathbf{v_k}(x(t_{k-1,s}), t_{k-1,s})\delta t + \frac{1}{2}\frac{d\mathbf{v_k}}{dt}\delta t^2 + O(\delta t^3) \right\|^2$$

$$= \|\mathbf{v_k}(x(t_{k-1,s}), t_{k-1,s})\|^2\delta t^2 + O(\delta t^3),$$

as we assume that $\mathbf{v_k}$ is $L$-Lipschitz continuous and it follows that $|\frac{d\mathbf{v_k}}{dt}| \le L$. The higher-order terms $O(\delta t^3)$ become negligible as $\delta t \to 0$.

Summing the expected squared displacements over all mini-intervals, we obtain:

$$\sum_{s=0}^{S-1} \mathbb{E} \left[ \|x(t_{k-1,s+1}) - x(t_{k-1,s})\|^2 \right] = \delta t^2 \sum_{s=0}^{S-1} \mathbb{E} \left[ \|\mathbf{v_k}(x(t_{k-1,s}), t_{k-1,s})\|^2 \right] + O\left(S \cdot \delta t^3\right).$$

Now, we examine the L.H.S. of Proposition 2 by approximating the integral of the expected squared velocity using a Riemann sum:

$$\int_{t_{k-1}}^{t_k} \mathbb{E}_{x(t)} \left[ \|\mathbf{v_k}(x(t), t)\|^2 \right] dt = \delta t \sum_{s=0}^{S-1} \mathbb{E} \left[ \|\mathbf{v_k}(x(t_{k-1,s}), t_{k-1,s})\|^2 \right] + O\left(S \cdot \delta t^2\right)$$

$$= \delta t \left[ \frac{1}{\delta t^2} \sum_{s=0}^{S-1} \mathbb{E} \left[ \|x(t_{k-1,s+1}) - x(t_{k-1,s})\|^2 \right] + O(S \cdot \delta t) \right] + O(S \cdot \delta t^2)$$

$$= \frac{1}{\delta t} \sum_{s=0}^{S-1} \mathbb{E} \left[ \|x(t_{k-1,s+1}) - x(t_{k-1,s})\|^2 \right] + O\left(S \cdot \delta t^2\right),$$

where the Riemann sum error term $O(S \cdot \delta t^2)$ arises from a well-known result (for instance, see Chapter 1 of Axler (2020)), given the assumption that $\mathbf{v_k}$ is $L-$Lipschitz continuous.

A.2   PROOFS IN SECTION 3.3

**Proposition 3.** (Objective Reformulation) *Denote* $h_k = t_k - t_{k-1}$, *and let* $\mathbf{s_k} = \nabla \log f_k(x)$ *denote the score function of* $f_k$. *As* $h_k \to 0$ *and with* $\gamma = \frac{1}{2}$ *(so that* $\alpha = \frac{S}{2h_k}$), *the objective in (10) becomes equivalent to the following:*

$$\min_{\mathbf{v_k} = \mathbf{v_k}(\cdot, 0)} \mathbb{E}_{x \sim f_{k-1}} \left[ -T_{f_k} \mathbf{v_k} + \frac{1}{2} \|\mathbf{v_k}\|^2 \right], \quad T_{f_k} \mathbf{v_k} := \mathbf{s_k} \cdot \mathbf{v_k} + \nabla \cdot \mathbf{v_k}.$$

*Proof:*

From the Neural ODE (1) and using Taylor's expansion, we obtain:

$$x(t_k) - x(t_{k-1}) = \int_{t_{k-1}}^{t_k} \mathbf{v_k}(x(s), s) ds = h_k \mathbf{v_k}(x(t_{k-1}), t_{k-1}) + O(h_k^2)$$

Next, by performing Taylor expansion of $\tilde{E}_k(x(t_k))$ around $t_{k-1}$:

$$\tilde{E}_k(x(t_k)) = \tilde{E}_k(x(t_{k-1})) + (x(t_k) - x(t_{k-1})) \nabla \tilde{E}_k(x(t_{k-1})) + O(h_k^2)$$
$$= \tilde{E}_k(x(t_{k-1})) + h_k \nabla \tilde{E}_k(x(t_{k-1})) \cdot \mathbf{v_k}(x(t_{k-1}), t_{k-1}) + O(h_k^2)$$

Besides, we also have that:

$$\int_{t_{k-1}}^{t_k} \nabla \cdot \mathbf{v_k}(x(s), s) ds = h_k \nabla \cdot \mathbf{v_k}(x(t_{k-1}), t_{k-1}) + O(h_k^2).$$

As $h_k \to 0$, we no longer need to divide the time interval, i.e., $S = 1$. By defining the score function as $\mathbf{s_k} = \nabla \log f_k = -\nabla \tilde{E}_k$, the objective function (10) can be then approximated as:

$$\mathbb{E}_{x \sim f_{k-1}} \left[ \tilde{E}_k(x(t_k)) - \int_{t_{k-1}}^{t_k} \nabla \cdot \mathbf{v_k}(x(s), s) \, ds + \frac{1}{2h_k} \|x(t_k) - x(t_{k-1})\|^2 \right]$$

$$= \mathbb{E}_{x \sim f_{k-1}} \left[ \left( \tilde{E}_k(x(t_{k-1})) - h_k \mathbf{s_k}(x(t_{k-1})) \cdot \mathbf{v_k}(x(t_{k-1}), t_{k-1}) + O(h_k^2) \right) \right.$$

$$\left. - \left( h_k \nabla \cdot \mathbf{v_k}(x(t_{k-1}), t_{k-1}) + O(h_k^2) \right) + \frac{1}{2h_k} \|h_k \mathbf{v_k}(x(t_{k-1})) + O(h_k^2)\|^2 \right]$$

$$= \mathbb{E}_{x \sim f_{k-1}} \left[ \tilde{E}_k(x) + h_k \left( -\mathbf{s_k}(x) \cdot \mathbf{v_k}(x, t_{k-1}) - \nabla \cdot \mathbf{v_k}(x, t_{k-1}) + \frac{1}{2} \|\mathbf{v_k}(x, t_{k-1})\|^2 \right) + O(h_k^2) \right]$$

Since $\mathbb{E}_{x(t_{k-1}) \sim f_{k-1}}[\tilde{E}_k(x(t_{k-1}))]$ is independent of $\mathbf{v_k}(x, t)$, as $h_k \to 0$, the minimization of the leading term is equivalent to:

$$\min_{\mathbf{v_k} = \mathbf{v_k}(\cdot, 0)} \mathbb{E}_{x \sim f_{k-1}} \left[ -T_{f_k} \mathbf{v_k} + \frac{1}{2} \|\mathbf{v_k}\|^2 \right], \quad T_{f_k} \mathbf{v_k} := \mathbf{s_k} \cdot \mathbf{v_k} + \nabla \cdot \mathbf{v_k}.$$

**Proposition 4:** (Optimal Velocity Field as Score Difference) *Suppose* $h_k \to 0$ *and* $\gamma = \frac{1}{2}$. *Let* $f_{k-1}$ *and* $f_k$ *be continuously differentiable on* $\mathbb{R}^d$. *Assume that* $\nabla \cdot \mathbf{v_k}(x)$ *exists for all* $x \in \mathbb{R}^d$, *and* $\nabla \cdot \mathbf{v_k}(x)$, $\mathbf{s_{k-1}}$ *and* $\mathbf{s_k}$ *belong to* $L^2(f_{k-1})$. *Assume that the components of* $\mathbf{v_k}$ *are independent and* $\lim_{\|x\| \to \infty} f_{k-1}(x) \|\mathbf{v_k}(x)\|_2 = 0$. *Under these conditions, the minimizer of (10) is:*

$$\mathbf{v_k}^* = \mathbf{s_k} - \mathbf{s_{k-1}}.$$

*Proof:*

Under the assumptions that $h_k \to 0$ and $\gamma = \frac{1}{2}$, we begin by considering the equivalent minimization objective derived in Proposition 3:

$$\min_{\mathbf{v_k}} J(\mathbf{v_k}) := \min_{\mathbf{v_k}} \mathbb{E}_{x \sim f_{k-1}} \left[ -T_{f_k} \mathbf{v_k} + \frac{1}{2} \|\mathbf{v_k}\|^2 \right], \quad T_{f_k} \mathbf{v_k} := \mathbf{s_k} \cdot \mathbf{v_k} + \nabla \cdot \mathbf{v_k}.$$

Expanding the objective functional, we have:

$$\mathbb{E}_{x \sim f_{k-1}}\left[-\mathbf{s_k} \cdot \mathbf{v_k} - \nabla \cdot \mathbf{v_k} + \frac{1}{2}\|\mathbf{v_k}\|^2\right] = \int_{\mathbb{R}^d} f_{k-1}(x)\left(-\mathbf{s_k}(x) \cdot \mathbf{v_k}(x) - \nabla \cdot \mathbf{v_k}(x) + \frac{1}{2}\|\mathbf{v_k}(x)\|^2\right)\,dx.$$

Define $B_r = \{x \in \mathbb{R}^d : \|x\| \leq r\}$, and let $\partial B_r$ denote the boundary of $B_r$, which is the sphere of radius $r$. Under the assumption that $\lim_{\|x\| \to \infty} f_{k-1}(x)\|\mathbf{v_k}(x)\|_2 = 0$, we have the following:

$$
\begin{aligned}
|\int_{\mathbb{R}^d} \nabla \cdot (f_{k-1}\,\mathbf{v_k})\,dx| &= \lim_{r \to \infty} |\int_{B_r} \nabla \cdot (f_{k-1}\mathbf{v_k})\,dx| \\
&= \lim_{r \to \infty} |\int_{\partial\{x \in \mathbb{R}^d : \|x\| < r\}} f_{k-1}(x)\mathbf{v_k}(x) \cdot \mathbf{n}(x)dS(x)| \\
&\leq \lim_{r \to \infty} \int_{\partial\{x \in \mathbb{R}^d : \|x\| < r\}} f_{k-1}\|\mathbf{v_k}\|_2\|\mathbf{n_k}\|_2 dS(x) \\
&= \lim_{r \to \infty} \int_{\partial\{x \in \mathbb{R}^d : \|x\| < r\}} f_{k-1}\|\mathbf{v_k}\|_2 dS(x) \\
&= 0
\end{aligned}
$$

Therefore, $\int_{\mathbb{R}^d} \nabla \cdot (f_{k-1}\,\mathbf{v_k})\,dx = 0$. Next, we further expand the divergence theorem:

$$
\begin{aligned}
0 &= \int_{\mathbb{R}^d} \nabla \cdot (f_{k-1}(x)\mathbf{v_k}(x))dx \\
&= \int_{\mathbb{R}^d} f_{k-1}(x)\nabla \cdot \mathbf{v_k}(x)dx + \int_{\mathbb{R}^d} \mathbf{v_k}(x) \cdot \nabla f_{k-1}(x)dx \\
&= \int_{\mathbb{R}^d} f_{k-1}(x)\nabla \cdot \mathbf{v_k}(x)dx + \int_{\mathbb{R}^d} \mathbf{v_k}(x) \cdot \mathbf{s_{k-1}}(x)\,f_{k-1}(x)\,dx
\end{aligned}
$$

Substitute the result back into the objective functional, we have:

$$
\begin{aligned}
\mathbb{E}_{x \sim f_{k-1}}\left[-\mathbf{s_k} \cdot \mathbf{v_k} - \nabla \cdot \mathbf{v_k} + \frac{1}{2}\|\mathbf{v_k}\|^2\right] &= \int_{\mathbb{R}^d} f_{k-1}(x)\left(-\mathbf{s_k}(x) \cdot \mathbf{v_k}(x) - \nabla \cdot \mathbf{v_k}(x) + \frac{1}{2}\|\mathbf{v_k}(x)\|^2\right)\,dx \\
&= \int_{\mathbb{R}^d} f_{k-1}(x)\left((\mathbf{s_{k-1}}(x) - \mathbf{s_k}(x)) \cdot \mathbf{v_k}(x) + \frac{1}{2}\|\mathbf{v_k}(x)\|^2\right)\,dx.
\end{aligned}
$$

The integrand does not involve $\nabla v_{k,j}(x), j = 1, \cdots d$ and higher-order derivatives. Assuming the components $v_{k,j}, j = 1, \cdots, d$ of $\mathbf{v_k}$ are independent, we can take the functional derivative component-wise and set them to zero:

$$\frac{\delta J}{\delta \mathbf{v_k}} = f_{k-1}\left(\mathbf{v_k} + (\mathbf{s_{k-1}} - \mathbf{s_k})\right) = 0,$$

Since $f_{k-1} > 0$ for all $x$, this implies:

$$\mathbf{v_k}^* = \mathbf{s_k} - \mathbf{s_{k-1}}.$$

A.3   PROOFS IN SECTION 5.2

**Density Ratio Estimation (DRE)** *By optimizing the following loss function:*

$$\mathcal{L}_k(\theta_k) = \mathbb{E}_{x(t_{k-1}) \sim f_{k-1}}\left[\log(1 + e^{-r_k(x_i(t_{k-1}))})\right] + \mathbb{E}_{x(t_k) \sim f_k}\left[\log(1 + e^{r_k(x_i(t_k))})\right],$$

*the model learns an optimal $r^*(x; \theta_k) = \log \frac{f_{k-1}(x)}{f_k(x)}$.*

*Proof:*

Express the loss function as integrals over $x$:

$$\mathcal{L}_k = \int f_{k-1}(x)\log\left(1 + e^{-r_k(x)}\right)\,dx + \int f_k(x)\log\left(1 + e^{r_k(x)}\right)\,dx.$$

Compute the functional derivative of $\mathcal{L}_k$ with respect to $r_k$:

$$\frac{\delta \mathcal{L}_k(r_k)}{\delta r_k} = -f_{k-1}(x) \cdot \frac{e^{-r_k(x)}}{1 + e^{-r_k(x)}} + f_k(x) \cdot \frac{e^{r_k(x)}}{1 + e^{r_k(x)}}.$$

Next, we can set the derivative $\delta l_k / \delta r_k(x)$ to zero to find the minimizer $r_k^*(x)$:

$$r_k^*(x) = \ln\left(\frac{f_{k-1}(x)}{f_k(x)}\right).$$

Therefore, by concatenating each $r_k^*(x)$, we obtain

$$r^*(x) = \sum_{k=1}^{K} r_k^*(x) = \log \frac{f_{K-1}(x)}{f_K(x)} \cdot \frac{f_{K-2}(x)}{f_{K-1}(x)} \cdot \ldots \cdot \frac{f_0(x)}{f_1(x)} = \log \frac{f_0(x)}{f_K(x)} = \log \frac{\pi_0(x)}{q^*(x)},$$

the log density ratio between $\pi_0(x)$ and $q^*(x)$.

## B    EQUIVALENCE TO WASSERSTEIN GRADIENT FLOW WHEN $\beta = 1$

In this section, we demonstrate the equivalence of Annealing Flow to the Wasserstein Gradient Flow when all $\beta_k, k = 1, 2, \ldots, K$, are set to 1, and when using a static Wasserstein regularization, instead of the dynamic Wasserstein regularization derived in Proposition 9.

*Langevin Dynamics and Fokker-Planck Equation:* Langevin Dynamics is represented by the following SDE.

$$dX_t = -\nabla E(X_t)\, dt + \sqrt{2}\, dW_t, \tag{16}$$

where $E$ is the energy function of the equilibrium density $f(x, T) = q(x)$. Standard generative model training typically focuses on the case of a normal equilibrium, i.e., $E(x) = \frac{x^2}{2}$ and $q(x) \propto e^{-E(x)}$. Let $X_0 \sim p_X$ and denote the density of $X_t$ by $\rho(x, t)$. The Langevin Dynamics also corresponds to the Fokker-Planck Equation (FPE), which describes the evolution of $\rho(x, t)$ towards the equilibrium $\rho(x, T) = q(x)$, as follows:

$$\partial_t \rho = \nabla \cdot (\rho \nabla E + \nabla \rho), \quad \rho(x, 0) = p_X(x). \tag{17}$$

In our algorithm, we focus on sampling from any distribution using its energy function, requiring only the unnormalized density. Therefore, $E(X_t)$ represents the potential of any target density $q(x)$. We initialize samples from an easy-to-sample distribution, $\rho(x, 0) = \pi_0(x)$, such as $N(0, I_d)$, and aim to learn the trajectory between $\pi_0(x)$ and the target $q(x)$. Therefore, sampling from $q(x)$ boils down to first drawing $x(0)$ from $\pi_0(x)$ and then moving $x(0)$ along the learned trajectory to finally obtain $x(T) \sim q(x)$.

*JKO Scheme:* The Jordan-Kinderlehrer-Otto (JKO) scheme (Jordan et al., 1998) is a time discretization scheme for gradient flows to minimize $\mathrm{KL}(\rho\|q)$ under the Wasserstein-2 metric. Given a target density $q$ and a functional $\mathcal{F}(\rho) = \mathrm{KL}(\rho\|q)$, the JKO scheme approximates the continuous gradient flow of $\rho(x, t)$ by solving a sequence of minimization problems. Assume there are $K$ steps with time stamps $0 = t_0, t_1, \cdots, t_K = T$, at each time stamp $t_k$, the scheme updates $\rho_k$ at each time step by minimizing the functional

$$\rho_k = \arg\min_{\rho}\left(\mathcal{F}(\rho) + \frac{1}{2\tau} W_2^2(\rho, \rho_{k-1})\right), \tag{18}$$

where $W_2(\rho, \rho_{k-1})$ denotes the squared 2-Wasserstein distance between the probability measures $\rho$ and $\rho_k$. It was proven in Jordan et al. (1998) that as $h = t_k - t_{k-1}$ approaches 0, the solution $\rho(\cdot, kh)$ provided by the JKO scheme converges to the solution of (17), at each step $k$.

The later works Xu et al. (2024a) have further shown that solving for the transport density $\rho_k$ by (18) is equivalent to solving for the transport map $\mathcal{T}_k$ by:

$$\mathcal{T}_k = \arg \min_{\mathcal{T}:\mathbb{R}^d \to \mathbb{R}^d} \left( KL(\mathcal{T}_{\#}\rho_{k-1}\|q) + \frac{1}{2\tau}\mathbb{E}_{x \sim \rho_{k-1}}\|x - \mathcal{T}_k(x)\|^2 \right) \tag{19}$$

Therefore, we immediately see that the Wasserstein gradient flow based on the discretized JKO scheme is equivalent to (6) when we set each $\tilde{f}_k(x)$ as the target distribution $q(x)$, i.e., when all the $\beta_k$ are set to 1, and when the second term in the objective (6) is relaxed to a static $W_2$ regularization.

This suggests that when the modes of the densities are not too far apart, and it is difficult to find a proper sequence of $\beta_k$, one can simply set all $\tilde{f}_k(x)$ in our algorithm as the target density $q(x)$, to construct a discretized sequence of transport maps based on Wasserstein gradient descent.

# C  EXPERIMENTAL DETAILS

## C.1  EVALUATION METRICS

To assess the performance of our model, we utilized two key metrics: Maximum Mean Discrepancy (MMD) and Wasserstein Distance, both of which measure the divergence between the true samples and the samples generated by the algorithms.

*Maximum Mean Discrepancy (MMD)*

MMD is a non-parametric metric used to quantify the difference between two distributions based on samples. Given two sets of samples $X_1 \in \mathbb{R}^{n_1 \times d}$ and $X_2 \in \mathbb{R}^{n_2 \times d}$, MMD computes the kernel-based distances between these sets. Specifically, we employed a Gaussian kernel:

$$k(x,y) = \exp\{-\alpha\|x - y\|_2^2\},$$

parameterized by a bandwidth $\alpha$. The MMD is computed as follows:

$$\text{MMD}(X_1, X_2) = \frac{1}{n_1^2}\sum_{i,j} k(X_1^i, X_1^j) + \frac{1}{n_2^2}\sum_{i,j} k(X_2^i, X_2^j) - \frac{2}{n_1 n_2}\sum_{i,j} k(X_1^i, X_2^j),$$

where $k(\cdot, \cdot)$ represents the Gaussian kernel. In our experiments, we set $\alpha = 1/\gamma^2$ and $\gamma = 0.1 \cdot \text{median\_dist}$, where median\_dist denotes the median of the pairwise distances between the two datasets.

*Wasserstein Distance*

In addition to MMD, we used the Wasserstein distance, which measures the cost of transporting mass between distributions. Given two point sets $X \in \mathbb{R}^d$ and $Y \in \mathbb{R}^d$, we compute the pairwise Euclidean distance between the points. The Wasserstein distance is then computed using the optimal transport plan via the linear sum assignment method (from scipy.optimize package):

$$W(X, Y) = \frac{1}{n}\sum_{i=1}^{n} \|X_{r(i)} - Y_{c(i)}\|_2,$$

where $r(i)$ and $c(i)$ are the optimal row and column assignments determined through linear sum assignment.

In all experiments, we sample 10,000 points from each model and generate 10,000 true samples from the GMM to calculate and report both MMD and Wasserstein distance. Note that the smaller the two metrics mentioned above, the better the sampling performance.

## C.2  HUTCHINSON TRACE ESTIMATOR

The objective functions in (10) and (11) involve the calculation of $\nabla \cdot \mathbf{v_k}(x, t)$, i.e., the divergence of the velocity field represented by a neural network. This may be computed by brute force using reverse-mode automatic differentiation, which is much slower and less stable in high dimensions.

We can express $\nabla \cdot \mathbf{v_k}(x, t) = \mathbb{E}_{\epsilon \sim N(0, I_d)} \left[ \epsilon^T J_v(x) \epsilon \right]$, where $J_v(x)$ is the Jacobian of $\mathbf{v_k}(x, t)$ at $x$. Given a fixed $\epsilon$, we have $J_v(x)\epsilon = \lim_{\sigma \to 0} \frac{\mathbf{v_k}(x + \sigma \epsilon) - \mathbf{v_k}(x)}{\sigma}$, which is the directional derivative of $\mathbf{v_k}$ along the direction $\epsilon$. Thus, for a sufficiently small $\sigma > 0$, we can propose the following estimator (Hutchinson, 1989; Xu et al., 2024a):

$$\nabla \cdot \mathbf{v_k}(x, t) \approx \mathbb{E}_{\epsilon \sim N(0, I_d)} \left[ \epsilon^T \frac{\mathbf{v_k}(x + \sigma \epsilon, t) - \mathbf{v_k}(x, t)}{\sigma} \right]. \tag{20}$$

This approximation becomes exact as $\sigma \to 0$. In our experiments, we set $\sigma = 0.02/\sqrt{d}$.

### C.3  OTHER ANNEALING FLOW SETTINGS

*Time stamps and numerical integration*

By selecting $K$ values of $\beta$, we divide the original time scale $[0, 1]$ of the Continuous Normalizing Flow (2) and (3) into $K$ intervals: $[t_{k-1}, t_k]$ for $k = 1, 2, \ldots, K$. Notice that the learning of each velocity field $\mathbf{v_k}$ depends only on the samples from the $(k-1)$-th block, not on the specific time stamp. Therefore, we can re-scale each block's time interval to $[0, 1]$, knowing that using the time stamps $[(k-1)h, kh]$ yields the same results as using $[0, 1]$ for the neural network $\mathbf{v_k}(x, t)$. For example, the neural network will learn $\mathbf{v_k}(x, 0) = \mathbf{v_k}(x, (k-1)h)$ and $\mathbf{v_k}(x, 1) = \mathbf{v_k}(x, kh)$, regardless of the time stamps.

Recall that we relaxed the shortest transport map path into a dynamic $W_2$ regularization loss via Proposition 2. This requires calculating intermediate points $x(t_{k-1,s})$, where $s = 0, 1, \ldots, S$. We set $S = 3$, evenly spacing the points on $[t_{k-1}, t_k]$, resulting in the path points $x(t_{k-1}), x(t_{k-1} + h_k/3), x(t_{k-1} + 2h_k/3), x(t_k)$. To compute each $x(t_{k-1,s})$, we integrate the velocity field $\mathbf{v_k}$ between $t_{k-1}$ and $t_{k-1,s}$, using the Runge-Kutta method for numerical integration. Additionally, for each $x(t_{k-1,s})$, we calculate the velocity field at an intermediate time step between $t_{k-1,s-1}$ and $t_{k-1,s}$ to enable accurate numerical integration. Specifically, to calculate $x(t+h)$ based on $x(t)$ and an intermediate time stamp $t + \frac{h}{2}$:

$$x(t + h) = x(t) + \frac{h}{6} \left( k_1 + 2k_2 + 2k_3 + k_4 \right),$$

$$k_1 = \mathbf{v}(x(t), t), \quad k_2 = \mathbf{v}\left( x(t) + \frac{h}{2}k_1, t + \frac{h}{2} \right),$$

$$k_3 = \mathbf{v}\left( x(t) + \frac{h}{2}k_2, t + \frac{h}{2} \right), \quad k_4 = \mathbf{v}\left( x(t) + hk_3, t + h \right)$$

Here, $h$ is the step size, and $\mathbf{v}(x, t)$ represents the velocity field.

*The choice of $\beta_k$*

In the experiments on Gaussian Mixture Models (GMM) and Exp-Weighted Gaussians with various dimensions and radii, we set the number of intermediate $\beta_k$ values to 8, equally spaced such that $\beta_0 = 0$, $\beta_1 = 1/8$, $\beta_2 = 2/8, \ldots, \beta_8 = 1$. We chose the easy-to-sample distribution $\pi_0(x)$ as $N(0, I_d)$. Finally, we added 2 refinement blocks. The intermediate distributions are defined as:

$$\tilde{f}_k(x) = \pi_0(x)^{1 - \beta_k} \tilde{q}(x)^{\beta_k}.$$

In the experiment on the Truncated Normal Distribution, we did not select $\beta_k$ in the same manner as for the GMM and Exp-Weighted Gaussian distributions. Instead, following the same Annealing philosophy, we construct a gradually transforming bridge from $\pi_0(x)$ to $\tilde{q}(x) = 1_{|x| \geq c} N(0, I_d)$ by setting each intermediate density as:

$$\tilde{f}_k(x) = 1_{\|x\| \geq c/(k+1)} N(0, I_d).$$

This choice also demonstrates that our Annealing Flow is highly flexible and capable of handling a wide range of challenging distributions.

In the experiment on funnel distributions, we set all $\beta_k = 1$. Therefore, as discussed in Appendix B, the algorithm becomes equivalent to a Wasserstein gradient descent problem. We also set the number of blocks to 8, consistent with the other experiments. This indicates that when the densities are largely concentrated in one region, one can simply set $\beta_k$ to 1 and use a few blocks to find the optimal transport path based on Wasserstein gradient descent.

*The objective*

During the experiments, we found that using the Taylor approximation (as described in Proposition 3, with a slight modification such that the expansion is around $x(t_k)$, allowing the loss to include the velocity field term): $\tilde{E}_k(x_{k-1}) - \tilde{E}_k(x_k) = (-h_k)\nabla E(x_k) \cdot \mathbf{v_k}$, and replacing the energy function $\tilde{E}_k(x_k)$ generally led to better performance. In our experiments on the GMM, Funnel distribution, and Exp-weighted Gaussian, we consistently used this form. For the experiments on the Truncated Normal and Bayesian Logistic Regression, the original $\tilde{E}_k(x_k)$ was used.

*Neural networks and selection of other hyperparameters*

The neural network structure in our experiments is consistently set with hidden layers of size 32-32-32. During implementation, we observed that when $d \leq 5$, even a neural network with a single hidden layer of size 32 can perform well for sampling. However, for consistency across all experiments, we uniformly set the structure to 32-32-32.

We sample 100,000 data points from $N(0, I_d)$ for training, with a batch size of 1,000. The Adam optimizer is used with a learning rate of 0.0001, and the maximum number of iterations for each block $\mathbf{v_k}$ is set to 1,000. An additional two blocks are added for refinement after $\beta_K = 1$.

Different numbers of test samples are used for reporting the experimental results: 5,000 points are sampled and plotted for the experiment on Gaussian Mixture Models, 5,000 points for the experiment on Truncated Normal Distributions, 10,000 points for the experiment on Funnel Distributions, and 10,000 points for the experiment on Exp-Weighted Gaussian with 1,024 modes in 10D space.

## C.4 BAYESIAN LOGISTIC REGRESSION

We use a hierarchical Bayesian structure for logistic regression across a range of datasets provided by LIBSVM. The detailed setting of the Bayesian Logistic Regression is as follows.

We adopt the same Bayesian logistic regression setting as described in Liu & Wang (2016), where a hierarchical structure is assigned to the model parameters. The weights $\beta$ follow a Gaussian prior, $p_0(\beta|\alpha) = N(\beta; 0, \alpha^{-1})$, and $\alpha$ follows a Gamma prior, $p_0(\alpha) = \text{Gamma}(\alpha; 1, 0.01)$. The datasets used are binary, where $x_i$ has a varying number of features, and $y_i \in \{+1, -1\}$ across different datasets. Sampling is performed from the posterior distribution:

$$p(\beta, \alpha|D) \propto Gamma(\alpha; 1, 0.01) \cdot \prod_{d=1}^{D} N(\beta_d; 0, \alpha^{-1}) \cdot \prod_{i=1}^{n} \frac{1}{1 + \exp(-y_i \beta^T x_i)},$$

We set $\beta_k = 1$ and use 8 blocks to train the Annealing Flow.

During testing, we use all algorithms to sample 1,000 particles of $\beta$ and $\alpha$ jointly, and use $\{\beta^{(i)}\}_{i=1}^{1000}$ to construct 1,000 classifiers. The mean accuracy and standard deviation are then reported in Table 2. Additionally, the average log posterior in Table 2 is reported as:

$$\frac{1}{|D_{\text{test}}|} \sum_{x,y \in D_{\text{test}}} \log \frac{1}{|C|} \sum_{\theta \in C} p(y|x, \theta).$$

## C.5 IMPORTANCE FLOW

We report the results of the importance sampler (discussed in Section 5) for estimating $\mathbb{E}_{x \sim N(0,I)} \left[ 1_{\|x\| \geq c} \right]$ with varying $c$ and dimensions, based on our Annealing Flow. To estimate $\mathbb{E}_{x \sim N(0,I)} \left[ 1_{\|x\| \geq c} \right]$, we know that the theoretically optimal proposal distribution which can achieve 0 variance is $\tilde{q}^*(x) = 1_{\|x\| \geq c} N(0, I)$. Then the estimator becomes:

$$\mathbb{E}_{X \sim \pi_0(x)} [h(X)] = \mathbb{E}_{X \sim q^*(x)} \left[ \frac{\pi_0(x)}{q^*(x)} \cdot h(x) \right] \approx \frac{1}{n} \sum_{i=1}^{n} \frac{\pi_0(x_i)}{q^*(x_i)} \cdot h(x_i), \quad x_i \sim q^*(x),$$

where $\pi_0(x) = N(0, I_d)$, $h(x) = 1_{\|x\| \geq c}$ and $q^*(x) = Z \cdot \tilde{q}^*(x)$.

Therefore, the Importance Flow consists of two parts: First, using Annealing Flow to sample from $\tilde{q}^*(x)$; second, constructing a Density Ratio Estimation (DRE) neural network using samples from $\{x_i\}_{i=1}^{n} \sim \tilde{q}^*(x)$ and $\{y_i\}_{i=1}^{n} \sim N(0, I_d)$, as discussed in Section 5.2. The estimator becomes:

$$\frac{1}{n} \sum_{i=1}^{n} DRE(x_i) \cdot h(x_i).$$

The Naive MC results comes from directly using $\{y_i\}_{i=1}^n \sim N(0, I_d)$ to construct estimator $\frac{1}{n} \sum_{i=1}^n 1_{\|y_i\| \geq c}$. When $c \geq 6$, the Naive MC methods consistently output 0 as the result.

In our experiment, we use a single DRE neural network to construct the density ratio between $\pi_0(x)$ and $q^*(x) = Z \cdot 1_{\|x\| \geq c} N(0, I)$ directly. The neural network structure consists of hidden layers with sizes 64-64-64. The size of the training data is set to 100,000, and the batch size is set to 10,000. We use 30 to 70 epochs for different distributions, depending on the values of $c$ and dimension $d$. The Adam optimizer is used, with a learning rate of 0.0001. The test data size is set to 1,000, and all results are based on 200 estimation rounds, each using 500 samples.

### C.6 DETAILS OF OTHER ALGORITHMS

The Algorithm 2, 3, and 4 introduce the algorithmic framework of Metropolis-Hastings (MH), Hamiltonian Monte Carlo (HMC), and Parallel Tempering (PT) compared in our experiments.

---

**Algorithm 2** Metropolis-Hastings Algorithm

1: Initialize $x_0$
2: **for** $t = 1$ to $N$ **do**
3: Propose $x^* \sim q(x^*|x_{t-1})$
4: Compute acceptance ratio $\alpha = \min\left(1, \frac{\pi(x^*)q(x_{t-1}|x^*)}{\pi(x_{t-1})q(x^*|x_{t-1})}\right)$
5: Sample $u \sim \text{Uniform}(0, 1)$
6: **if** $u < \alpha$ **then**
7:  $x_t = x^*$
8: **else**
9:  $x_t = x_{t-1}$
10: **end if**
11: **end for**
12: **return** $\{x_t\}_{t=0}^N$

---

**Algorithm 3** Hamiltonian Monte Carlo (HMC)

1: Initialize $x_0$
2: **for** $t = 1$ to $N$ **do**
3: Sample $p \sim \mathcal{N}(0, M)$
4: Set $(x, p) \leftarrow (x_{t-1}, p)$
5: **for** $i = 1$ to $L$ **do**
6:  $p \leftarrow p - \frac{\epsilon}{2} \nabla U(x)$
7:  $x \leftarrow x + \epsilon M^{-1} p$
8:  $p \leftarrow p - \frac{\epsilon}{2} \nabla U(x)$
9: **end for**
10: Compute acceptance ratio $\alpha = \min\left(1, \exp(H(x_{t-1}, p_{t-1}) - H(x, p))\right)$
11: Sample $u \sim \text{Uniform}(0, 1)$
12: **if** $u < \alpha$ **then**
13:  $x_t = x$
14: **else**
15:  $x_t = x_{t-1}$
16: **end if**
17: **end for**
18: **return** $\{x_t\}_{t=0}^N$

---

In our experiments, we set the proposal density as $q(x'|x) = \mathcal{N}(x; 0, I_d)$. We use 5 replicas in Parallel Tempering (PT), with a linear temperature progression ranging from $T_1 = 1.0$ to $T_{\max} = 2.0$, and an exchange interval of 100 iterations. For HMC, we set the number of leapfrog steps to 10, with a step size ($\epsilon$) of 0.01, and the mass matrix $M$ is set as the identity matrix. Additionally, we use the default hyperparameters as specified in SVGD (Liu & Wang, 2016), MIED (Li et al., 2023), and AI-Sampler (Egorov et al., 2024). In the actual implementation, we found that the time required for SVGD to converge increases significantly with the number of samples. Therefore, in most experiments, we sample 1000 data points at a time using SVGD, aggregate the samples, and then generate the final plot.

## D MORE RESULTS

We adopt the standard Annealing Flow framework discussed in this paper for experiments on Gaussian Mixture Models (GMM), Truncated Normal distributions, and Exp-Weighted Gaussian distributions. For experiments on funnel distributions, we set each $\tilde{f}_k(x)$ as the target $q(x)$, under which the Annealing Flow objective becomes equivalent to the Wasserstein Gradient Flow based on the JKO scheme, as discussed in B. Please refer to C.3 for $\beta_k$ selections.

---

**Algorithm 4** Parallel Tempering Algorithm

---

1: Initialize replicas $\{x_1, x_2, \ldots, x_{\text{num\_replicas}}\}$ with Gaussian noise
2: Initialize temperatures $\{T_1, T_2, \ldots, T_{\text{num\_replicas}}\}$
3: **for** $i = 1$ to iterations **do**
4:    **for** $j = 1$ to num\_replicas **do**
5:       Propose $x_j^* \sim q(x_j^*|x_j)$ {Using Metropolis-Hastings step for each replica}
6:       Compute acceptance ratio $\alpha_j = \frac{\pi(x_j^*)}{\pi(x_j)}$
7:       Sample $u \sim \text{Uniform}(0, 1)$
8:       **if** $u < \alpha_j$ **then**
9:          $x_j = x_j^*$
10:       **end if**
11:       Store $x_j$ in samples for replica $j$
12:    **end for**
13:    **if** $i \mod \text{exchange\_interval} = 0$ **then**
14:       **for** $j = 1$ to num\_replicas $- 1$ **do**
15:          Compute energies $E_j = -\log(\pi(x_j) + \epsilon)$, $E_{j+1} = -\log(\pi(x_{j+1}) + \epsilon)$
16:          Compute $\Delta = \left(\frac{1}{T_j} - \frac{1}{T_{j+1}}\right)(E_{j+1} - E_j)$
17:          Sample $u \sim \text{Uniform}(0, 1)$
18:          **if** $u < \exp(\Delta)$ **then**
19:             Swap $x_j \leftrightarrow x_{j+1}$
20:          **end if**
21:       **end for**
22:    **end if**
23: **end for**
24: **return** samples from all replicas

---

*Gaussian Mixture Models (GMM)*

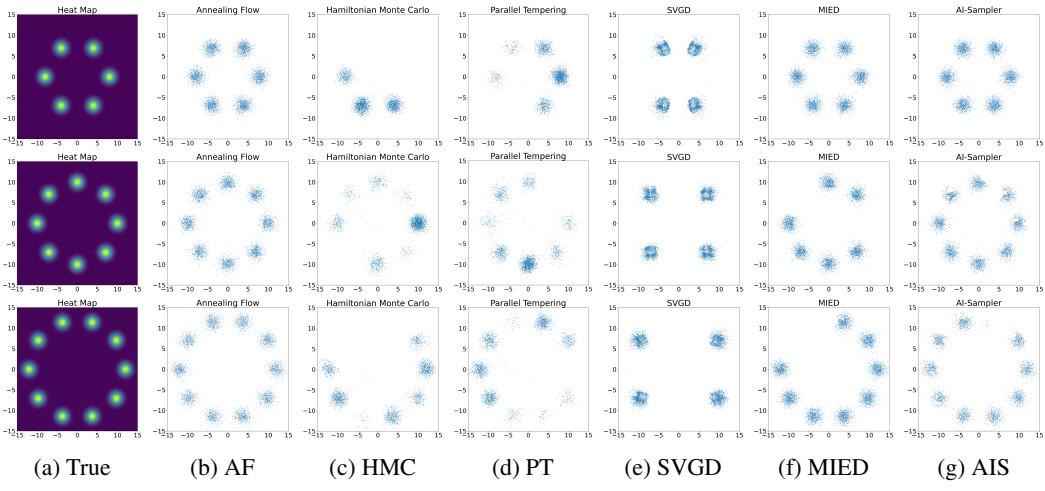

(a) True     (b) AF     (c) HMC     (d) PT     (e) SVGD     (f) MIED     (g) AIS

Figure 7: Sampling methods for Gaussian Mixture Models (GMM) with 6, 8, and 10 modes distributed on circles with radii $r = 8, 10, 12$.

*Evaluation Metrics:* We report 1) the Maximum Mean Discrepancy (MMD) and 2) the Wasserstein Distance for the GMM experiments, as both metrics require access to true data samples. The results for these metrics are presented in Table 4. Please refer to C.1 for more details.

Table 4: MMD and Wasserstein Distance results: $\cdot/\cdot$ represents MMD/Wasserstein. The first row corresponds to $d = \{$dimension$\}$ GMM-$\{$Number of Modes$\}$.

| | $d = 2$ GMM-8 | $d = 2$ GMM-12 | $d = 3$ GMM-8 | $d = 4$ GMM-16 | $d = 5$ GMM-32 | $d = 6$ GMM-64 |
|---|---|---|---|---|---|---|
| AF | **2.32E-03/7.38E-01** | **3.01E-03/8.05E-01** | **5.82E-03/1.97E+00** | **1.25E-03/3.33E+00** | **1.57E-03/2.82E+00** | **4.31E-03/3.53E+00** |
| HMC | 7.33E-02/6.28E+00 | 9.06E-02/8.73E+00 | 9.92E-02/1.12E+01 | 9.76E-02/1.98E+01 | 2.14E-01/2.53E+01 | 2.15E-01/3.03E+01 |
| PT | 6.27E-02/5.71E+00 | 9.01E-02/7.91E+00 | 8.83E-02/1.07E+01 | 8.98E-02/1.53E+01 | 1.18E-01/1.83E+01 | 1.05E-01/2.13E+01 |
| SVGD | 9.35E-02/9.97E+00 | 1.85E-01/1.82E+01 | 9.81E-02/1.13E+01 | 9.63E-02/2.07E+01 | 1.98E-01/2.45E+01 | 1.32E-01/2.34E+01 |
| MIED | **2.34E-03**/8.01E-01 | 6.28E-03/9.35E-01 | 8.01E-02/2.52E+00 | 3.88E-02/0.89E+01 | 9.88E-03/7.89E+00 | 2.03E-02/1.13E+01 |
| AIS | **2.33E-03**/7.92E-01 | 4.02E-03/8.13E-01 | 7.55E-02/2.38E+00 | 5.26E-03/5.53E+00 | 6.37E-03/3.83E+00 | 1.87E-02/9.73E+00 |

*Truncated Normal Distribution*

Relaxations are applied to the Truncated Normal Distribution in all experiments except for MH, HMC, and PT. Specifically, we relax the indicator function $1_{\|x\|\geq c}$ to $\frac{1}{1+\exp(-k(\|x\|-c))}$. We set $k = 20$ for all experiments. AIS is designed for continuous densities, and we similarly relax the densities in SVGD and MIED, following the approach used in AF. The resulting plots are as follows:

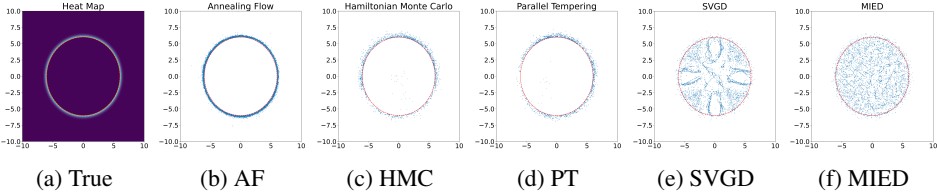

(a) True  (b) AF  (c) HMC  (d) PT  (e) SVGD  (f) MIED

Figure 8: Sampling Methods for Truncated Normal Distributions with Radius $c = 6$, together with the failure cases of SVGD and MIED.

Each algorithm draws 5,000 samples. It can be observed that MCMC-based methods, including HMC and PT, produce many overlapping samples. This occurs because when a new proposal is rejected, the algorithms retain the previous sample, leading to highly correlated sample sets.

Table 5: Proportion of Annealing Flow Samples Within c, Across Different Dimensions

| Proportion Within c | $c = 4$ | $c = 6$ | $c = 8$ |
|---|---|---|---|
| $D = 2$ | 0.17% | 0.18% | 1.78% |
| $D = 3$ | 0.20% | 0.23% | 3.23% |
| $D = 4$ | 0.68% | 1.48% | 3.68% |
| $D = 5$ | 1.46% | 3.37% | 4.12% |
| $D = \mathbf{10}$ | 2.13% | 4.68% | 7.13% |

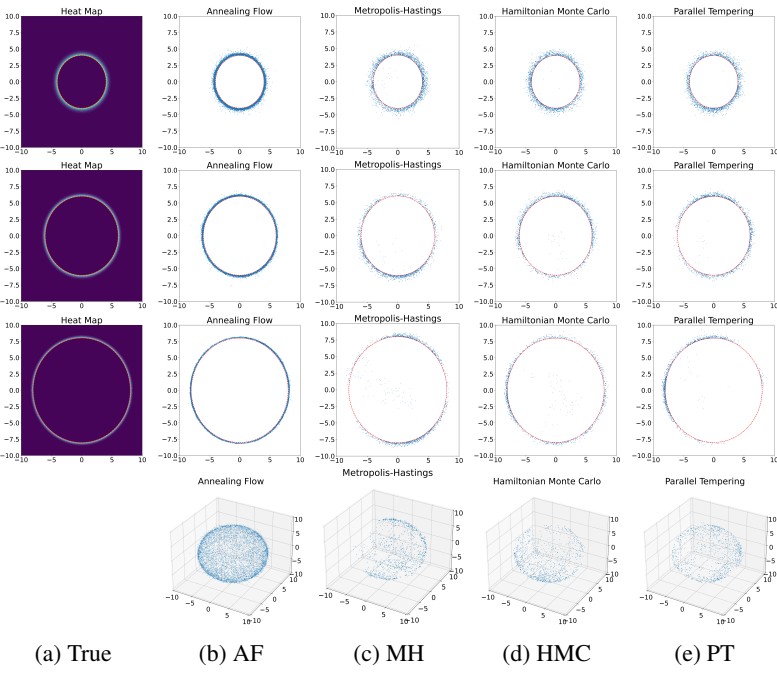

(a) True  (b) AF  (c) MH  (d) HMC  (e) PT

For dimensions $d > 2$, visualizing the results by comparing the sample positions using a red sphere surface becomes challenging. Therefore, we calculate the proportion of samples within radius $c$. A lower proportion indicates better sampling performance. Table 5 presents these results. We also calculate the proportion of the

surface $\|x\| = c$ covered by the samples for AF, MH, HMC, and PT. In all experiments with the Truncated Normal distribution, AF covers more than 95% of the surface area. However, when $d \geq 3$ and $c \geq 6$, all other methods cover less than 70% of the surface area.

*Funnel Distribution*

In the main paper, we present the sampling methods for the funnel distribution with $d = 5$, projected onto a 3D space. To assess the sample quality, here we present the corresponding results projected onto a 2D space, plotted alongside the density heat map.

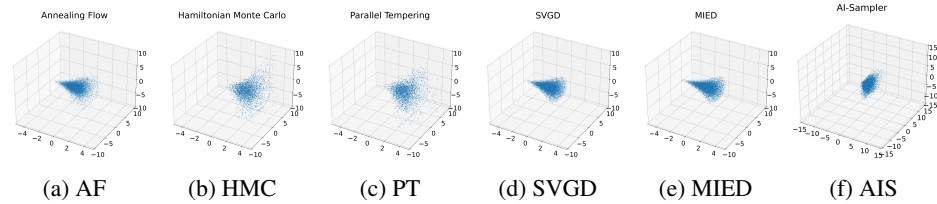

(a) AF     (b) HMC     (c) PT     (d) SVGD     (e) MIED     (f) AIS

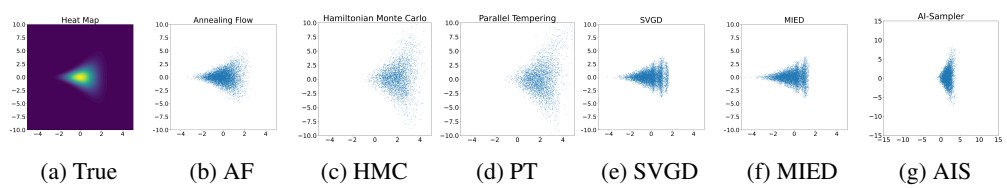

(a) True     (b) AF     (c) HMC     (d) PT     (e) SVGD     (f) MIED     (g) AIS

Figure 11: Sampling Methods for Funnel Distribution with $\sigma^2 = 0.81$ in Dimension $d = 5$, projected onto a $d = 3$ Space.

As seen from both figures, our AF method achieves the best sampling performance on the funnel distribution, while other methods, such as MIED and AIS, fail to capture the full spread of the funnel's tail. Additionally, PT, SVGD, and AIS all fail to capture the sharp part of the funnel's shape.

*Exp-Weighted Gaussian*

In the main paper, we present the sampling methods for the Exp-Weighted Gaussian distribution with 1024 modes in a **50D** space, projected onto a 3D space. To better assess the sample quality, we now present the corresponding results projected onto 2D and 1D spaces, plotted alongside the heat map and the true density, respectively.

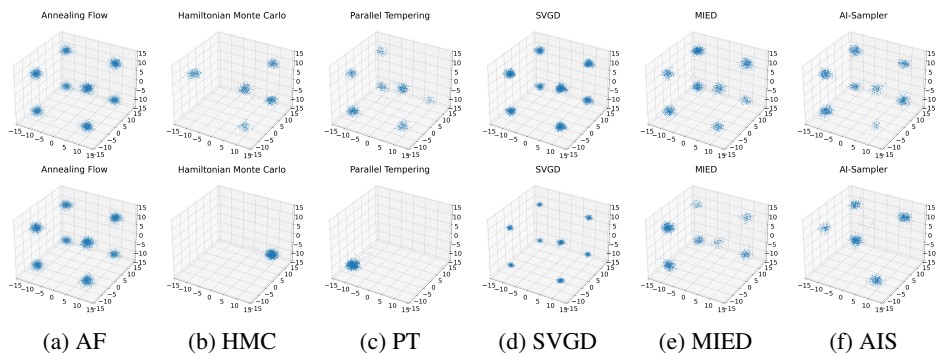

(a) AF     (b) HMC     (c) PT     (d) SVGD     (e) MIED     (f) AIS

Figure 12: Sampling Methods for an Exp-Weighted Gaussian Distribution with 1024 modes in **10D** (Top) and **50D** (Bottom), projected onto a 3D Space.

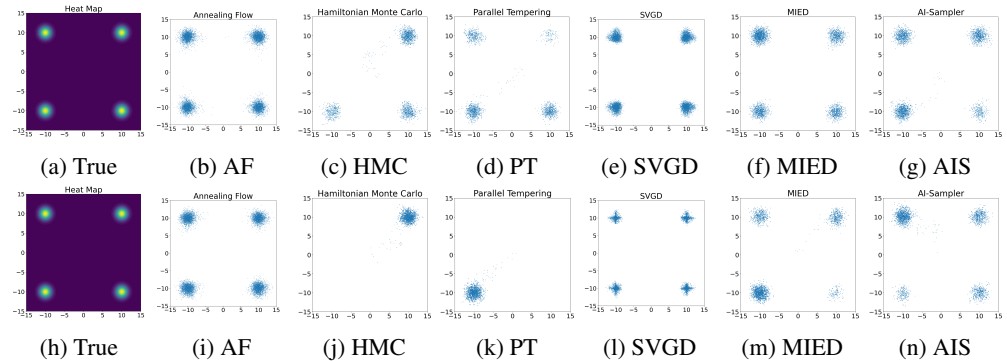

Figure 13: Sampling Methods for an Exp-Weighted Gaussian Distribution with 1024 modes in **10D** (Top) and **50D** (Bottom), projected onto a 2D Space.

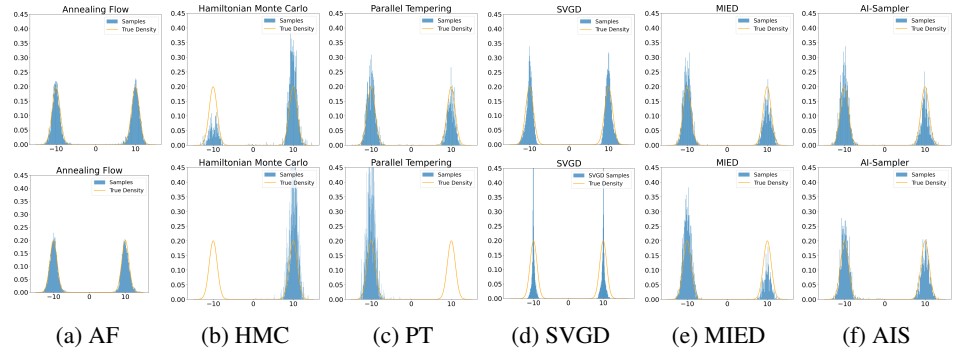

Figure 14: Sampling Methods for an Exp-Weighted Gaussian Distribution with 1024 modes in **10D** (Top) and **50D** (Bottom), projected onto a 1D Space.

As seen in Figures 13 and 14, AF produces balanced samples, and its 1D projection closely matches the true density. While both SVGD and MIED captured around 800 to 900 modes, their samples across the modes are imbalanced, as observed in the figures. We projected onto each dimension, and the results were similar.

## D.1 COMPARISONS

Table 6: Comparisons of Different Sampling Methods

| Method | Key Characteristics | Advantages | Disadvantages |
|---|---|---|---|
| **Annealing Flow (AF)** | - Continuous Normalizing Flow-based approach. 
 - Leverages annealing principles for sampling challenging high-dimensional, multi-modal distributions. 
 - Uses transport maps to transform samples from a base distribution to the target distribution. | - Independent sampling. 
 - Balanced mode exploration. 
 - Handles multi-modal distributions effectively. 
 - Once trained, the sampling process is very fast 
 - Scales linearly with sample size and dimensionality. | - Requires pre-training, which can be computationally expensive. |
| **MCMC** | - Metropolis-Hastings, Parallel Tempering, Hamiltonian Monte Carlo (HMC) variants. 
 - Samples sequentially from the target distribution, with each sample depending on the previous one. | - Flexible, general-purpose. 
 - Doesn't require pre-training. | - Slow mixing time. 
 - Struggles with multi-modal distributions. 
 - Sample correlation reduces effective sample size (ESS). 
 - Imbalanced mode exploration. |
| **Particle-Based Optimization (SVGD, MIED)** | - Relies on particle dynamics and kernel methods to sample from the target distribution. | - No burn-in period. 
 - Less sample correlation than MCMC. 
 - Encourages global search. | - Kernel computations scale polynomially with sample size. 
 - Sensitive to kernel hyperparameters. |
| **NN-Assisted MCMC** | - Uses neural networks to accelerate or guide MCMC methods. 
 - Combines the expressive power of neural networks with MCMC. | - Can speed up the explorations of MCMC methods. 
 - Leverages NN for improved sampling efficiency. | - Inherits some limitations of MCMC, such as slow mixing, correlated samples, and mode imbalance. |
| **Score-based Diffusion** | - Learns score functions to iteratively perturb samples towards the target distribution. | - Strong theoretical guarantees for sampling specific distributions. | - Limited generalization to arbitrary distributions, as score functions are analytically derived. 
 - Challenging in complex, high-dimensional distributions |

Annealing Flow (AF) requires pre-training, typically taking 10-20 minutes for tasks with dimensions $< 10$, and around 30 minutes for tasks around dimension 50. For 50D experiments, training a single $v_k$ with a neural network structure of 32-32-32 and 1000 gradient steps takes approximately 2–3 minutes. Once trained, AF samplers are very efficient: generating 10,000 samples in just 1.5 seconds. These pre-trained samplers can be reused at any time, offering significant speed advantages. In contrast, MCMC methods, such as Metropolis-Hastings or Hamiltonian Monte Carlo, require about 1 minute to sample 10,000 points, and their performance deteriorates in high-dimensional, multi-modal settings. Moreover, particle-based methods, like SVGD, struggle significantly when generating more than 3,000 samples, requiring about 20 minutes for that many samples. Therefore, we believe that users can take advantage of AF's offline training, as it allows the samplers to be trained once and then efficiently reused for sampling whenever needed.

## D.2 IMPORTANCE FLOW

The importance flow discussed and experimented with in this paper requires a given form of $\pi_0(x)$, and thus, a given form of $\tilde{q}^*(x) = \pi_0(x) \cdot |h(x)|$ for estimating $\mathbb{E}_{X \sim \pi_0(x)}[h(X)]$. In our experimental settings, $\tilde{q}^*(x) = 1_{\|x\| \geq c} N(0, I_d)$ can be regarded as the Least-Favorable-Distribution (LFD). We conducted a parametric experiment for the case where $\tilde{q}^*(x)$ has the given analytical form.

However, we believe future research may extend this approach to a distribution-free model. That is, given a dataset without prior knowledge of its distribution, one could attempt to learn an importance flow for sampling from its Least-Favorable Distribution (LFD) while minimizing the variance. For example, in the case of sampling from the LFD and obtaining a low-variance IS estimator for $P_{x \sim \pi(x)}(\|x\| \geq c)$, one may use the following distribution-free loss for learning the flow:

$$\min_\theta \frac{1}{n} \sum_{i=1}^n \left[ 1\{\mathcal{T}(x_i; \theta) \leq c\} \cdot \|\mathcal{T}(x_i; \theta) - c\|^2 \right] + \gamma \int_0^1 \|\mathbf{v}(x(t), t; \theta)\|^2, \tag{21}$$

where the first term of the loss pushes the dataset $\{x_i\}_{i=1}^n$ towards the Least-Favorable tail region, while the second term ensures a smooth and cost-optimal transport map. Note that the above loss assumes no prior knowledge of the dataset distribution $\pi(x)$ or the target density $q(x)$.

Xu et al. (2024b) has also explored this to some extent by designing a distributionally robust optimization problem to learn a flow model that pushes samples toward the LFD $Q^*$, which is unknown and learned by the model through a risk function $\mathcal{R}(Q^*, \phi)$. Such framework has significant applications in adversarial attacks, robust hypothesis testing, and differential privacy. Additionally, the recent paper by Ribera Borrell et al. (2024) introduces a dynamic control loss for training a neural network to approximate the importance sampling control. We believe that by designing an optimal control loss in line with the approaches of these two papers, one can develop a distribution-free Importance Flow for sampling from the LFD of a dataset while minimizing the variance of the adversarial loss, which can generate a greater impact on the fields of adversarial attacks and differential privacy.

