# OpenReview forum: "Annealing Flow Generative Model Towards Sampling High-Dimensional and Multi-Modal Distributions"
_ICLR.cc/2025/Conference — ICLR 2025 Conference Withdrawn Submission_

### Official Review · Reviewer_HDGZ · 2024-10-29

**Soundness:** 1
**Presentation:** 2
**Contribution:** 2
**Rating:** 1
**Confidence:** 3

**Summary:**

This paper addresses the challenge of sampling from unnormalized probability distributions by introducing a novel approach called Annealed Flow (AF). The authors define a sequence of annealed distributions and propose learning continuous normalizing flows between each distribution in the sequence. Furthermore, the paper presents an additional method, termed Importance Flow (IF), which allows for computing expectations by first sampling the optimal importance distribution using AF and then learning the associated importance ratio. Experimental results validate the effectiveness of AF and IF across a diverse set of target distributions.

**Strengths:**

- The importance flow technique is original

**Weaknesses:**

- The paper is poorly written (and follows unusual notations for instance with the normalizing constants)
- The paper suffers from theoretical issues/typos (KL between unnormalised distributions, proof of proposition 1, ...)
- The authors don't mention or compare to direct competitors [1, 2] who solve the same problem with the same tools
- Poor experimental part with very few comments

**Questions:**

- Could you compare your work against [1] and [2] which have solved the same problem with very similar tools ?
- Could you rewrite sections 2 and 3 but defining the normalizing constants from the beginning ? I think it will make the paper much clearer. Moreover, it would avoid writing KL divergences between unnormalised distributions as done extensively in section 3 and appendix 1.
- I suspects mistakes have been made in the proof of proposition 1. Could you explain the following points ?
1. In common litterature, the Kullback-Leiber divergence is defined between normalized distributions as $D(p || q) = \mathbb{E}_{X \sim p}[\log p(X) / q(X)]$. However, L760 shows a completely different definition (different order and unormalized densities);
2. According to the beginning of section 3.2, the quantity $\rho_k$ is the density of the flow following $v_k$ which means that $\log \rho_k(x(t_{k-1}))$ is not independent of $v_k(x(s),s)$  which would contradict L800.
- The block-wise training procedure (Sec. 4.1) uses AF samples from step $k-1$ to train for step $k$. However, those samples are approximate and have no reasons to be calibrated according to $f_{k-1}$. Can we trust those samples to compute expectations with respect the $f_{k-1}$ ? (Note that the same remark goes for the Importance Flow procedure)
- At L260, you claim that your training procedure only requires a single neural network. Doesn't your sampling procedure require to keep all the intermediates $\{v_k\}_{k = 1}^K$ ?
- Could you perform the experiment on Fig. 1 with differently weighted modes ? I feel like recovering the weights of the modes is the true challenge.
- Could you provide numerical comparisons with recent VI approaches [3,4,5] and with similar approaches [1,2] ? Those methods are important competitors.
- Could you compute the metrics given in [6] which seem computable in your case ? Those metrics really focus on multimodal distributions.
- Could you provide metrics's mean and standard deviation (as well as the number of samples used) for each experiment ?
- Could you display true samples from the Funnel distribution alongside yours ? It would clarify the pros and cons of each algorithm.


[1] Tian, Y., Panda, N., & Lin, Y. (2024). Liouville Flow Importance Sampler. In Proceedings of the 41st International Conference on Machine Learning (pp. 48186–48210). PMLR.

[2] Fan, M., Zhou, R., Tian, C., & Qian, X. (2024). Path-Guided Particle-based Sampling. In Proceedings of the 41st International Conference on Machine Learning (pp. 12916–12934). PMLR.


[3] Francisco Vargas, Will Sussman Grathwohl, & Arnaud Doucet (2023). Denoising Diffusion Samplers. In The Eleventh International Conference on Learning Representations .

[4] Qinsheng Zhang, & Yongxin Chen (2022). Path Integral Sampler: A Stochastic Control Approach For Sampling. In International Conference on Learning Representations.

[5] Richter, L., & Berner, J. (2024). Improved sampling via learned diffusions. In International Conference on Learning Representations.

[6] Blessing, D., Jia, X., Esslinger, J., Vargas, F., & Neumann, G. (2024). Beyond ELBOs: A Large-Scale Evaluation of Variational Methods for Sampling. In Proceedings of the 41st International Conference on Machine Learning (pp. 4205–4229). PMLR.

---

> ### Author Response · Authors · 2024-11-13
> **Reply to Reviewer HDGZ**
>
> We appreciate your detailed comments and suggestions and will revise our draft accordingly. We would also like to clarify some of your questions related to our draft:
>
> * Could you compare your work against [1] and [2] which have solved the same problem with very similar tools?
>
> We have read the two papers carefully, and noted that their methods are based on gradient flows, instead of Annealing-guided flow with Wasserstein regularization. Through experiments, we observed that without $W_{2}$-regularization, the paths become less organized and more dispersed, which means the flow often misses the optimal path; and if the density modes are widely separated or in high-dimensional spaces, without Annealing-guidance, the gradient flow often fails. These two key features help ensure success in high-dimensional multi-modal cases (an example is in Figure 6, a density with 1024 modes in 50D). We will develop theories on the guiding effects of $W_{2}$ and Annealing. Thank you for identifying these two papers; we will include comparison experiments with theirs.
>
> * Could you rewrite sections 2 and 3 but defining the normalizing constants from the beginning? I think it will make the paper much clearer. Moreover, it would avoid writing KL divergences between unnormalised distributions as done extensively in section 3 and appendix 1.
>
> Thank you for your suggestions. We will revise accordingly.
>
> * In common litterature, the Kullback-Leiber divergence is defined between normalized distributions as $D(p \parallel q) = \mathbb{E}_{X \sim p} \left[ \log \frac{p(X)}{q(X)} \right]$. However, L760 shows a completely different definition.
> * According to the beginning of section 3.2, the quantity $\rho_k$ is the density of the flow following $v_k$, which means that $\log \rho_k(x(t_{k-1}))$ is not independent of $v_k(x(s), s)$, which would contradict L800.
>
> We're sorry for the confusion; there is a typing error in the proof. We should have written the first step as $KL(T f_{k-1} \parallel f_k) = E_{X\sim \rho(t_{k})}\left[ \log T f_{k-1}(X) / f_{k}(X) \right]$, where $\rho(t)$ represents the density evolution governed by equation (2).
>
> Here, $\rho(t_{k}) = T f_{k-1}$, and from the constraints in (3), we see that the density evolution is subject to the equations (1) and (2). Then subject to (1) and (2), we can write the objective $E_{X\sim \rho(t_{k})}\left[ \log T f_{k-1}(X) / f_{k}(X) \right] = E_{X\sim \rho(t_{k-1})}\left[ \log T f_{k-1}(X(t_{k})) / f_{k}(X(t_{k})) \right]$, where $\log T f_{k-1}(X(t_{k})) =\log \rho(X(t_{k})) = \log \rho(X(t_{k-1})) - \int_{t_{k-1}}^{t_k} \nabla \cdot v_k(X(s), s) ds$.
>
> For the second question: $v_{k}$ flows density from $\rho(X(t_{k-1}))$ to $\rho(X(t_{k}))$, where $X(t_{k}) = X(t_{k-1}) +\int_{t_{k-1}}^{t_{k}} v_k(X(s), s) ds$. Therefore, $E_{X\sim \rho(t_{k-1})} [ \log \rho(X(t_{k-1})) ]$ is a constant that is independent of $v_{k}$. (It depends on $v_{1},...,v_{k-1}$.) We are sorry again for the typing typo that leads to the misunderstanding.
>
> * At L260, you claim that your training procedure only requires a single neural network. Doesn't your sampling procedure require to keep all the intermediates
>
> Our algorithm adopts a block-wise training procedure instead of end-to-end training. Our algorithm keeps the intermediate $v_{k}$. However, during training, only one NN (one $v_{k}$) is trained at each time, thus greatly enhancing training efficiency.
>
> * Could you perform the experiment on Fig. 1 with differently weighted modes ? I feel like recovering the weights of the modes is the true challenge.
>
> Thank you for the suggestion. After submission, we implemented modes with different weights, and the results are also convincing. We will include them in our future submission.
>
> * Could you compute the metrics given in [6] which seem computable in your case ? Those metrics really focus on multimodal distributions.
> * Could you provide metrics's mean and standard deviation (as well as the number of samples used) for each experiment ?
> * Could you display true samples from the Funnel distribution alongside yours ? It would clarify the pros and cons of each algorithm.
>
> Thank you! We will.
>
> We wish to conclude by highlighting that we are the first to use Annealing-guided CNFs with $W_{2}$ regularization, which are the key to efficient explorations of all modes in high-dimensional spaces. Our experiments focus more on extreme distributions (high-dimensional densities with a large number of widely separated modes), which are distinct from many other CNFs studies. In our recent draft, we proved that the infinitesimal optimal velocity field represents the score difference between two consecutive densities. We will continue refining the theories, and thank you for these useful suggestions.

---

> > ### Comment · Reviewer_HDGZ · 2024-11-23
> >
> > I thank the authors for their prompt and developed answer to the reviewers questions and I am eager to see the additional experiments. However, I think there are still many points to be discussed.
> >
> > I do agree that this paper is the first to use Annealing-guided CNFs with $W_2$ regularization. However, I still believe that it is very close to existing works. Given a sequence of unormormalized densities, this works suggests to learn multiple CNF to move between time $k$ and time $k+1$ by minimizing the following loss (Eq. 11) at time $k+1$ given previous samples
> >
> > $\mathcal{L}\_{k+1}(\hat{v}\_{k+1}) = \mathbb{E}\_{X\_k \sim f\_k} \left[-(\nabla \cdot \hat{v}\_{k+1})(X\_k) - \nabla \log f\_{k+1}(X\_k) \cdot \hat{v}\_{k+1}(X\_k) + \frac{1}{2}\lVert \hat{v}\_{k+1}(X\_k)\rVert^2 \right]$
> >
> > in an iterative fashion. On the other hand [1] suggests to learn a velocity field inducing an ODE with the same marginals as the sequence of densities by iterating the following loss
> >
> > $\mathcal{L}\_{k+1}(\hat{v}\_{k+1}) = \mathbb{E}\_{X\_{k+1} \sim f\_{k+1}} \left[\lVert(\nabla \cdot \hat{v}\_{k+1})(X\_{k+1}) + \nabla \log f\_{k+1}(X\_{k+1}) \cdot \hat{v}\_{k+1}(X\_k) + \left(\log \tilde{f}\_{k+1}(X\_{k+1}) - \mathbb{E}\_{Y\_{k+1} \sim f\_{k+1}}\left[\log \tilde{f}\_{k+1}(Y\_{k+1})\right]\right)\rVert^2\right]$
> >
> > They sample $X_{k+1} \sim f_{k+1}$ by doing importance sampling using samples from step $k$ as proposal. The two losses are extremely similar. Moreover, [1] has to use importance sampling in its loss to ensure that the expectation is taken with respect to the right measure while Eq. 11 blindly trust samples from the previous flow as samples from $f_k$ (What is the relative weight between the modes of $f_k$ and $f_{k+1}$ are different as it is often the case when using tempering paths ?). I think comparing to [1] and [2] from a theoretical and practical perspective is really mandatory. I am also very surprised that this work doesn't have to perform any importance sampling (unlike [1] and [2]) as it seems like an unavoidable problem as you are minimizing divergences between densities with unknown normalizing constants. Note that I believe that this is a general issue and is not specific to discrete flows unlike what the authors said in answer to reviewer **7EWK**. I believe that the announced experiments with unequally weights mixtures as well as the ablation study without the regularization term will clarify these points.
> >
> > I do agree with reviewers **FN4T** and **JJiD** that comparison with state-of-the-art sampling VI methods should be done (beyond Ai-sampler which is slightly orthogonal to the current literature). I would be especially interested in the comparison with diffusion based methods [3,4,5] (Or [6] for a review) which significantly improved upon NF techniques.
> >
> > I will be happy to increase my score upon the release of the new experiments-oriented revision.

---

### Official Review · Reviewer_C6rd · 2024-10-31

**Soundness:** 2
**Presentation:** 2
**Contribution:** 3
**Rating:** 5
**Confidence:** 4

**Summary:**

A continuous normalizing flow-based transport map guided by annealing is designed to sample from high-dimensional
and multi-modal distributions with unknown normalization factor. The approach supports effective exploration
of modes in high-dimensional spaces. The paper contains several comparison with state-of-the-art methods.
The method can be extended to a distribution-free model that allows
to learn an importance flow from a dataset for sampling its least-favorable distribution with
minimal variance.

**Strengths:**

The basic idea is straightforward, but interesting and the numerical results are quite convincing.

**Weaknesses:**

The mathematical notation is in many parts cursorily or wrong as the proof of Proposition 1.
The numerical experiments which are restricted to symmetric multimodal targets must be improved.
See questions and experimental suggestions  below.

**Questions:**

- in the introduction in connection with normalizing flows I am missing stochastic normalizing flows, see
\\
H. Wu, J. Köhler, and F. Noe, Stochastic normalizing flows, in Advances in Neural Information
Processing Systems 33, 5933–5944, 2020.
\\
P. Hagemann, J. Hertrich, G. Steidl,
   Stochastic normalizing flows for inverse problems: a Markov Chains viewpoint
SIAM Journal on Uncertainty Quantification 10 (3), 1162-1190,2022

- in (3): don't call the \textbf{value} of the integral $\mathcal T$; later this is your OT transport map.
- line 126: hint to Fig. 1 is too early here since $t_1,t_2$  is not explained so far
- line 152: ,,pushes density from $f_{k-1}(x)$ to $f_{k}(x)$, since these are function values, please skip the $x$;
this appears also at other places (but is a minor remark)
- line 159: I would not say that (3) is equivalent to (6)
- in (6) and in the following: what is the pushforward of $\tilde{f}_{k-1}$ by $\mathcal{T}$  for an unnormalized density? Definition of KL for
unnormalized densities ?; see Prop 1
The authors should write the definitions down, then they would see that this notation does not work.
- In general the authors switch between $f_k$ and $\rho_k$

- Prop 1 (Appendix A): Despite that the tilde notation does not work  and that $T$ is indeed $\mathcal T$ the authors have to correct
  line 770: first equality is wrong; you need here and in the following $\mathbb E_{x \sim \mathcal T _\#  \tilde f_{k-1} }$ (unfortunately the system does not translates this latex formula correctly, but the authors hopefully see what I mean);
 - from 774 to 776 is wrong and becomes correct with my correction above.
- First equality in 794 is wrong and in the final equality it should be $\log \rho_{k-1} (x(t_{k-1})) - \int ...$

- Prop. 2 is folklore; in formula (9) in the expectation value $x(t) \sim ??$ is missing

- Prop 3: Do not start ,,By Taylor expansion''. This is already the proof.
Write the correct assumptions on x and $\tilde E$ for this Taylor expansion.
Indeed 1. and 2. are folklore and there is nothing to prove.
However, the authors  wrongly replaced $x$ by $X$ in part 2 of the appendix.
- from formula (19) to (20) there is nothing to prove; in general the heuristic App. B appears superfluous to me,
but maybe it could be of interest for people not directly working in the field.
- The results of the experiments seem to depend heavily on the symmetry of the target density, Please show a modified experiment of Fig. 6 where you shift the target density in the first dimension e.g. by 5 to the left (and keep the variance of the latent distribution).
I guess that the reconstruction misses modes.
- the comparison with HMC is a little unfair. Please redo the HMC experiment with a different chain for each sample starting in the same latent distribution as your model. I would expect that for your symmetric target distribution this woks fine.

---

> ### Author Response · Authors · 2024-11-19
> **Reply to Reviewer C6rd**
>
> Thank you very much for your detailed review and kind suggestions on improvements! We acknowledge that the submitted draft was prepared in haste and contained some typographical errors and unclear notations, particularly in the proof of Proposition 1. We have carefully revised Proposition 1 and will upload a revised version shortly. Additionally, we would like to address some of your questions:
>
> * in the introduction in connection with normalizing flows I am missing stochastic normalizing flows
>
> Thank you for bringing such works to our attention. We will do literature review carefully and include the representative works of stochastic normalizing flows appropriately.
>
> * All your kind suggestions related to the proof of Propositions 1,2, and 3:
>
> We have addressed all the points and sincerely thank you for your detailed review and valuable comments.
>
> * The results of the experiments seem to depend heavily on the symmetry of the target density, Please show a modified experiment of Fig. 6 where you shift the target density in the first dimension e.g. by 5 to the left (and keep the variance of the latent distribution). I guess that the reconstruction misses modes.
>
> Thank you for your comment! After submission, we conducted additional experiments on densities with differently weighted modes and will include these in our future submissions. Additionally, we will incorporate experimental comparisons with other normalizing flow methods as suggested by other reviewers, and try to include more metrics.
>
> * the comparison with HMC is a little unfair. Please redo the HMC experiment with a different chain for each sample starting in the same latent distribution as your model. I would expect that for your symmetric target distribution this works fine.
>
> Our current HMC experiment initializes the chain from the same latent distribution, N(0,I), as used in our model. We followed the pseudo-code outlined in Algorithm 3 in Appendix, running a single continuous Markov chain where each sample $x_{t}$ depends on the location of the previous sample $x_{t-1}$. We are concerned that starting a different chain for each sample from N(0,I), would break the Markov property and detailed balance of Markov chain. Besides, we observed that in high-dimensional spaces, despite our efforts in adjusting the momentums and increasing leapfrog steps, HMC often fails to take $x_{t}$ to a new mode starting from $x_{t-1}$, as there is too much space to explore in 50D and the chain can hardly explore on the correct direction.
>
> We will carefully review the implementations of different algorithms and include additional experiments and comparisons in our future submissions. Once again, we sincerely thank you for your detailed review and thoughtful suggestions. We will upload a revised version of draft here very soon and hope it addresses at least some of your concerns. We will also continue refining the draft in preparation for future submissions.

---

### Official Review · Reviewer_JJiD · 2024-11-03

**Soundness:** 3
**Presentation:** 2
**Contribution:** 2
**Rating:** 3
**Confidence:** 5

**Summary:**

The present paper proposes a sampler based on a continuous normalizing flow. The idea is to decompose the transport into sub-transport tasks following an annealing scheme between a simple base distribution and the target. Velocity fields are learned independently for each such transport task so as to minimize a Kullback-Leibler divergence with a transport cost penalization. Once the networks are trained, the sampling is operated by simply sampling from the continuous NF.

The paper also points that the method can be employed to learn a flow as a proposal distribution in a rare event  importance sampling (IS) scenario. It is proposed that the IS weights are estimate by training a neural network to estimate the density ratio between the flow and the target.

Numerical experiments report a good mode exploration and qualitative good coverage for the proposed method compared to diverse samplers not relying on deep learning.

**Strengths:**

- The proposed methods addresses the important point of mode coverage in the task of sampling multimodal distributions in high dimension, which is demonstrated in numerical experiments.

**Weaknesses:**

- The paper lacks a related works section. Namely, the connections to  NN-assisted MCMCs, which are in my opinion the most related methods, are not discussed and the performances of these approaches are not reported. In my experience, these methods have better performance than what the paper seems to indicate. In particular, they typically do correct for the mode imbalance that a traditional MCMC based on local updates could not do.

- In this regard, the novelty of the paper is lesser than what it appears. Annealed Flow Transport  (Arbel et al 2021 in the paper) proposes a very related strategy.

- Limitations are not properly discussed:
	- The proposed sampling methods lacks theoretical guarantees of the accuracy of the sampling conversely to NN-assisted (Markov Chain) Monte Carlo samplers, such as Neural IS [3], Flow MC (Gabrie et al 2021, 2022 in the paper) or Annealed Flow Transport (Arbel et al 2021 in the paper).
	- The cost of training only one velocity field at the time is not discussed, nor how to choose the number of these intermediary steps.
	- For the importance flow method as well, the paper claims “The estimator is unbiased and can achieve zero variance theoretically” but this is only true if the density estimation ratio is predicted exactly by the neural network.



Minor:
- About “score based sampling that do not rely on neural networks” (line 66), the paper should also cite RDMC [1] and SLIPS [2].



[1] Xunpeng Huang and Hanze Dong and Yifan HAO and Yian Ma and Tong Zhang. Reverse Diffusion Monte Carlo. The Twelfth International Conference on Learning Representations, 2024, https://openreview.net/forum?id=kIPEyMSdFV

[2] Grenioux, Louis, Maxence Noble, Marylou Gabrié, and Alain Oliviero Durmus. “Stochastic Localization via Iterative Posterior Sampling.” In Proceedings of the 41st International Conference on Machine Learning, 16337–76. PMLR, 2024. https://proceedings.mlr.press/v235/grenioux24a.html.

[3] Müller, Thomas, Brian Mcwilliams, Fabrice Rousselle, Markus Gross, and Jan Novák. “Neural Importance Sampling.” ACM Transactions on Graphics 38, no. 5 (October 31, 2019): 1–19. https://doi.org/10.1145/3341156.

**Questions:**

- Is the condition of optimality of the transport crucial to the success of the method?

- I am surprised by PT’s result in Figure 2. I would expect that this exact sampler designed to sample from multimodal distribution works perfectly in this 1d example. Can the author justify underwhich circumstances the imbalance result was obtained?

- Table 3, what are the expected ground truth values?

- The notations between $\tilde f_k$ (the target annealing path of distributions)  and $f_k$ (the intermediary push forwards of the CNF) is maybe inconsistent in some places. For instance in (8) and (10), should the expectation be over $\tilde f_{k-1}$ instead of $f_{k-1}$ ?

---

> ### Author Response · Authors · 2024-11-13
> **Reply to Reviewer JJiD**
>
> We appreciate your comments and suggestions. We would like to clarify some weaknesses and questions you are concerned about:
>
> * The paper lacks a related works section. Namely, the connections to NN-assisted MCMCs, which are in my opinion the most related methods, are not discussed and the performances of these approaches are not reported. In my experience, these methods have better performance than what the paper seems to indicate. In particular, they typically do correct for the mode imbalance that a traditional MCMC based on local updates could not do.
>
> In the second-to-last paragraph, we discussed recent NN-assisted MCMC methods, including those that use neural networks as transition kernels and MCMCs based on normalizing flows. Notably, the "AI-Sampler" compared in our experiments is an NN-assisted MCMC method, which leverages NN to construct the transition kernels. We will include additional experimental comparisons with other NN-assisted methods, including "Neural Importance Sampling" you mentioned and the papers mentioned by Reviewer 7EWK and HDGZ.
>
> A unique advantage of our algorithm is that, unlike most NN-assisted sampling methods, it does not involve MCMC as part of the training process. This approach offers several benefits, including avoiding the long mixing times of MCMC (which can increase exponentially with dimensionality) and producing independent samples. NN-methods that lack Annealing-guidance and $W_{2}$ regularization often suffer from mode-seeking behaviors and often fail with densities that have widely-separated modes (like in the settings of our experiments).
>
> * In this regard, the novelty of the paper is lesser than what it appears. Annealed Flow Transport (Arbel et al 2021 in the paper) proposes a very related strategy.
>
> Thank you for pointing out this work. We noted that it focuses on discrete normalizing flows (NFs), which inherently exhibit mode-seeking behavior, rather than continuous normalizing flows (CNFs). To address such behavior, their paper uses SMC and importance weights to adjust the loss. Additionally, the experiments presented are primarily focused on scenarios with a low number of modes. Moreover, the mode-seeking tendencies of discrete NFs can become more pronounced as the number of target modes increases, and the SMC procedure may fail in high-dimensional density with separated modes, as in our experiment with 1024 widely separated modes in 50D space.
>
> Besides, through experiments, we observed that without $W_{2}$-regularization (as in their NFs), the transport paths become less organized and more dispersed, which means the flow often misses the optimal path. WWe are currently working on developing theoretical insights into the convergence properties of Annealing guidance and $W_{2}$ regularization.
>
> Thank you for your comment. We will make the comparisons clearer in our future submission.
>
> * The proposed sampling methods lacks theoretical guarantees of the accuracy of the sampling conversely to NN-assisted (Markov Chain) Monte Carlo samplers, such as Neural IS [3], Flow MC (Gabrie et al 2021, 2022 in the paper) or Annealed Flow Transport (Arbel et al 2021 in the paper).
>
> Our algorithm does not rely on an MCMC process, so fewer theoretical properties, such as the law of large numbers, CLT, and variance properties, can be proved. Nonetheless, after submission, we provided an official proof of the infinitesimal optimal velocity field, demonstrating it as the score difference between two consecutive annealing densities. We will further expand on the convergence properties of annealing guidance and $W_{2}$ regularization, which are key to our algorithm’s ability to explore the entire space effectively. Thanks for bringing us attention to develop more theories!
>
> * The cost of training only one velocity field at the time is not discussed, nor how to choose the number of these intermediary steps.
>
> Thank you for your detailed review. We will discuss these in the revised draft.
>
> * I am surprised by PT’s result in Figure 2. I would expect that this exact sampler designed to sample from multimodal distribution works perfectly in this 1d example. Can the author justify underwhich circumstances the imbalance result was obtained?
>
> For PT to balance samples between the modes, it must allow transitions between different modes frequently enough at different temperatures. When the two modes are far-separated, particles can have difficulty moving between them even at a high-temperature.
>
>  * Table 3, what are the expected ground truth values?
>
> This is shown in the first line of the table.
>
> * For the importance flow method as well, the paper claims “The estimator is unbiased and can achieve zero variance theoretically” but this is only true if the density estimation ratio is predicted exactly by the neural network.
>
> Thank you again for all these comments! We will revise our draft accordingly.

---

> > ### Comment · Reviewer_JJiD · 2024-11-19
> >
> > I thank the authors for responding to my comments.
> >
> > I will consider to revise my rating if the draft is revised. In particular, if they provide a comparison to Annealed Flow Transport (Arbel et al, 2021).

---

### Official Review · Reviewer_7EWK · 2024-11-03

**Soundness:** 2
**Presentation:** 3
**Contribution:** 1
**Rating:** 3
**Confidence:** 4

**Summary:**

This paper proposes to fit continuous normalizing flows (CNFs) to multi-modal unnormalized target distributions using annealing and optimal transport techniques. The proposed method first defines a sequence of annealed distributions interpolating the target distribution and a simple base distribution. Leveraging the dynamic optimal transport objective, the authors propose to train a sequence of CNFs to model the transition between each pair of neighboring annealed distributions. This method encourages the model to explore modes without relying on any target samples or running MCMC chains. Various experiments show that the proposed method has comparable or better performance than some of the traditional MCMC, particle-based and NN-based methods.

**Strengths:**

1. The paper is generally well-written. The description of the proposed method is clear and easy to follow.
2. The proposed sampler is able to explore modes in multi-modal target distributions, which is a desired property in many applications.
3. Once the model is trained, the sampling cost is low since it does not require running MCMC chains and sampling scales linearly with the sample size and dimensionality.
4. The proposed method is evaluated on various experimental settings and has comparable or better results than some of the tradition MCMC, particle-based and NN-based samplers, showing its good applicability.

**Weaknesses:**

1. The manuscript contains a GitHub link in Line 369-370, which reveals the authors' identity.

2. The proposed method lacks technical novelty and is a bit incremental, since it directly applies an optimal transport loss with KL relaxation to train a sequence of CNFs to model the transitions between neighboring annealed distributions, which is a trivial combination of existing techniques.

3. The scalability and reliability of the proposed training algorithm is questionable, since it requires training K seperate CNFs sequentially, each modeling the transition between one of the K neighboring pairs of annealed distributions. Furthermore, within each transition, the interval is further discretized into S grid points in order to estimate the integral.
- If K is small, the transition will be difficult and the model can fail to capture some modes. If S is small, the discretization error in numerical integration will be large and the gradient for updating the model will be inaccurate. If K and S are large, the proposed training algorithm will be very expensive as it requires backpropagating through all these steps and hierarchies.
- The approximation error in each transition will accumulate as the sampler gets closer to the target distribution due to the bootstrap-style training method.

4. It is also unclear how the proposed method should be positioned among other flows-based or NN-based samplers with similar techniques, since many important recent related works are not discussed in the paper, including
- flow-based sampler trained with annealing: [1, 2, 3]
- score/diffusion-based sampler: [4, 5, 6, 7, 9]
- optimal control-based sampler: [8]

5. Several closely related baselines are missing in the experiment section:
- At the moment, most baselines are MCMC/particle-based samplers.
- Since the proposed method is a flow-based model trained with the annealing technique, it should be empirically compared to at least the following models which use similar flow-annealing techniques: AFT [1], CRAFT [2], FAB [3].
- In addition, since the proposed method is broadly an NN-based sampler, ideally it should also be compared with some of other types of SOTA NN-based samplers, such as iDEM [6], DDS [7], PIS [8], PDIS [9].

6. For the toy 1D experiment in Figure 2, it would be more convincing if
- the two modes in the target distribution are unbalanced (i.e., one has a larger density than the other), as suggested by [5] to test whether the proposed method exhibits a similar blindeness issue as in some of the score-based samplers.
- it also includes results of other flow and annealing-based baselines in this comparison to get a sense of how these similar methods compare to each other, since they are more related to the proposed method in this paper.

7. [Optional] The proposed method is only tested on synthetic problems. Perhaps for future work, it would be nice if the authors could consider some real-world problems, such as sampling from Boltzmann distributions (i.e., Boltzmann generator, see e.g., [3, 6]) which share exactly the same problem setting as in this paper.

**Reference**:

[1] Arber et al., “Annealed flow transport Monte Carlo”, ICML 2021

[2] Matthews et al., “Continual repeated annealed flow transport Monte Carlo" ICML 2022.

[3] Midgley et al., “Flow annealed importance sampling bootstrap”, ICLR 2023

[4] Huang et al., “Reverse diffusion Monte Carlo”, ICLR 2024

[5] Chen et al., “Diffusive Gibbs sampling”, ICML 2024

[6] Akhound-Sadegh et al., “Iterated denoising energy matching for sampling from Boltzmann densities”, ICML 2024

[7] Vargas et al., “Denoising diffusion samplers”, ICLR 2023

[8] Zhang et al., “Path integral sampler: a stochastic control approach for sampling”, ICLR 2022

[9] Phillips et al., “Particle denoising diffusion sampler”, ICML 2024

**Questions:**

1. How sensitive is the weight hyperparameter $\gamma$ that controls the balance between the two terms in the dynamic optimal transport loss? Does the value of $\gamma$ vary a lot across different experiments? How the values of $\gamma$ were chosen in the experiments?

2. How does the hyperparameters $K$ (the number of intermediate distributions) and $S$ (the number of discretization grids within each transition between two intermediate distributions) scale with the dimensionality and multi-modaility of the target distribution?

3. [Typo] What is "MEID" in table 2? Is it "MIED"?

**Details Of Ethics Concerns:**

The manuscript contains a GitHub link in Line 369-370, which reveals the authors' identity.

---

> ### Author Response · Authors · 2024-11-19
> **Reply to Reviewer 7EWK**
>
> Thank you for your detailed review and these kind suggestions. We want to reply to some of your concerns:
>
> * The manuscript contains a GitHub link in Line 369-370
>
> We're sorry for the oversight, but we believe that the current GitHub link does not directly reveal authors' information (The link and the Annealing Flow repository does not contain any name or institution of authors). We will be very cautious in future submissions and will not include any link.
>
> * The proposed method lacks technical novelty and is a bit incremental, since it directly applies an optimal transport loss with KL relaxation to train a sequence of CNFs to model the transitions between neighboring annealed distributions, which is a trivial combination of existing techniques.
> * The scalability and reliability of the proposed training algorithm is questionable, since it requires training K separate CNFs sequentially, each modeling the transition between one of the K neighboring pairs of annealed distributions. Furthermore, within each transition, the interval is further discretized into S grid points in order to estimate the integral.
>
> Thank you for these comments. We wish to comment that Annealing principle is the key to ensure smooth transitions and efficient explorations of widely-separated density modes, especially in high-dimensional spaces, as demonstrated in our numerical experiments, which distinct our algorithm from others that also rely on Normalizing Flows. Without annealing procedures, the existing NF methods may easily fail when it comes to densities with widely separated modes.
>
> Besides, during experiments, we observed that without $W_{2}$-regularization (as in the most recent NF works), the transport paths become less organized and more dispersed, which means the flow often misses the optimal path. We are currently working on developing theories on the convergence properties of Annealing guidance and regularization.
>
> Indeed, more blocks (i.e., annealing densities) are required as the number of modes increases and becomes more separated. However, as mentioned in the last paragraph of Section 4.2, the training of AF can be performed offline, requiring only a single training process. Once trained, it can be deployed for sampling anytime as needed. In contrast, methods like MCMC and particle-based approaches often fail regardless of the training duration and must be reimplemented each time sampling is required. Additionally, recent normalizing flow (NF) works without annealing or $W_2$ regularization frequently encounter mode collapse in high-dimensional settings.
>
> Across all of our experiments, the discretization step $S$ is consistently set to 3, i.e., two intermediate $x(t_s)$ are used for numerical integration.
>
> * It is also unclear how the proposed method should be positioned among other flows-based or NN-based samplers with similar techniques, since many important recent related works are not discussed in the paper
> * Several closely related baselines are missing in the experiment section
>
> Thank you for your comments. We will include more experimental comparisons with other algorithms.  We noted that these Annealing works focus on discrete normalizing flows (NFs), which inherently exhibit mode-seeking behavior, rather than continuous normalizing flows (CNFs). To address such behavior, they use MC procedures and importance weights to correct the loss. Additionally, the experiments presented are primarily focused on scenarios with a low number of modes. Moreover, the mode-seeking tendencies of discrete NFs can become more pronounced as the number of target modes increases, and the SMC procedure may fail in high-dimensional density with separated modes, as in our experiment with 1024 widely separated modes in 50D space.
>
> Thank you for these suggestions. We will include clearer comparisons in future submission.
>
> * For the toy 1D experiment in Figure 2, it would be more convincing if ...
>
> Since the submission, we have conducted additional experiments on densities with differently weighted modes, which we will include in future submissions. We have also updated Figure 2 in our latest draft, which will be uploaded shortly after. However, we need more time to incorporate the results of other flow methods.
>
> Thank you once again for your detailed review and suggestions!

---

### Official Review · Reviewer_FN4T · 2024-11-04

**Soundness:** 3
**Presentation:** 2
**Contribution:** 3
**Rating:** 6
**Confidence:** 2

**Summary:**

The paper proposed a so-called Annealing flow (AF) sampling method for  high-dimensional multi-modal distributions.  The key idea is to learn a continuous normalizing flow (CNF)  based transport map, guided by annealing process, to transition samples from an easy-to-sample distribution to the target distribution. The feature of  AF training does not rely on samples from the target distribution. The results in dimensional up to 50  show that AF ensures effective and balanced mode exploration, especially for multi-model distribution by comparing  AF to other SOTA methods though various challenging distributions, particularly in high-dimensional and multi-modal settings, including  AF’s potential for sampling the least favorable distributions.

**Strengths:**

The paper present a continuous Normalizing flow  based method for sampling from challenging high dimensional multi-modal distributions.  There are some theoretically insights on the proposed models using the  predefined intermediate and the numerical results show the superiority of the proposed sampling method under variety of criteria.

**Weaknesses:**

1. The details of training process is missing. Especially, the cost of the training process of CNF flow is not mentioned,  and also how many intervals are needed in order to obtain a stable sampling process. It seems also crucial especially how the overall sampling results depend on the flow interval.

2. It is unclear how the method can be generalized to real high dimensional problems arising from real applications and distribution free problems where only samples are available.

3. The sample efficiency (both sampling speed and quality) is not commented and compared with different methods.

**Questions:**

1. The author should explain  the detail of training  including how the minimization problem is solved and how the parameters would affect the sampling performance.
2. Comments or preliminary results  on how it can appplied to any real high dimensional applications (>>50), for example a real physical machine learning problem.
3. Other comparisons results in terms of computational cost.

---

> ### Author Response · Authors · 2024-11-19
> **Reply to Reviewer FN4T**
>
> Thank you for these comments! We would like to reply to some of your questions and concerns:
>
> * The details of training process is missing. Especially, the cost of the training process of CNF flow is not mentioned, and also how many intervals are needed in order to obtain a stable sampling process. It seems also crucial especially how the overall sampling results depend on the flow interval.
> * The sample efficiency (both sampling speed and quality) is not commented and compared with different methods.
>
> Thank you for pointing these out. The costs of training and sampling are discussed in Appendix D.1, and the choice of intermediate annealing densities is discussed in Appendix C.3. We will include more details and discussions on these aspects in our future submissions.
>
> * It is unclear how the method can be generalized to real high dimensional problems arising from real applications and distribution free problems where only samples are available.
>
> The algorithm presented in this paper is designed for parametric statistical sampling in high-dimensional, multi-modal settings, a fundamental and important task in statistical fields such as Bayesian analysis and statistical physics. We have included experiments on hierarchical Bayesian logistic regression across a range of datasets.
>
> The task discussed in this paper focuses on statistical sampling from parametric distributions. We acknowledge that exploring data-driven scenarios—i.e., cases where only samples are available and the parametric form of the data densities is unknown—is indeed an interesting direction. In this work, we introduced a preliminary framework called Importance Flow for sampling rare regions of distributions and constructing low-variance estimators for rare probabilities. Additionally, we discussed the potential for extending this framework into a data-driven method, which we plan to focus on in future work.
>
> Thank you once again for your advice! We will revise the manuscript carefully based on your suggestions. In future submissions, we will include more detailed comparisons of training and sampling costs, as well as additional comparison experiments with other algorithms, as suggested by other reviewers.

---

### Author Response · Authors · 2024-11-19
**Reply to all reviewers**

The authors sincerely thank all the reviewers for their detailed reviews and thoughtful suggestions. We acknowledge that the submission was prepared in haste, leading to some typos in the mathematical notations, particularly in the proof of Proposition 1, which caused misunderstandings about our algorithm. We have now carefully revised the notations in the proof, and reviewers may refer to the latest version of the submission for clarification.

Additionally, we have included formal proofs for the properties of the infinitesimal optimal velocity field, $\mathbf{v_k}^*=\mathbf{s_k}-\mathbf{s_{k-1}}$, which were previously stated without proof in Section 3.3 (now provided in the Appendix of revised version).

As suggested by some reviewers, we acknowledge the need to include more comparison experiments, particularly with recent algorithms that use Normalizing Flows (NFs) for sampling. We still need additional time on these experiments, and we plan to reflect them in future submissions.

After a thorough literature review, We wish to emphasize that our work is the first to use Annealing-guided continuous normalizing flows (CNFs) with $W_2$ regularization, which are the key to efficient explorations of all modes in high-dimensional spaces. Unlike discrete NF algorithms, which often suffer from mode collapse issues, our approach demonstrates more effective performance on extreme distributions—high-dimensional densities with numerous widely separated modes. On the other hand, we will continue refining the theories, especially on the convergence and guidance-effects of Annealing and $W_2$ regularization.

Thank you once again for your efforts!

---

### Note · Authors · 2024-11-23

I have read and agree with the venue's withdrawal policy on behalf of myself and my co-authors.